**EMBO** *reports*

# Targeted protein degradation in *Escherichia coli* using CLIPPERs

Matylda Anna Izert-Nowakowska [ID][1], Maria Magdalena Klimecka[1], Anna Antosiewicz[1], Karol Wróblewski [ID][2], Jakub Józef Kowalski [ID][1], Katarzyna Justyna Bandyra[1], Tomasz Góral [ID][3], Sebastian Kmiecik[2], Remigiusz Adam Serwa[4] & Maria Wiktoria Górna [ID][1✉]

## Abstract

New, universal tools for targeted protein degradation in bacteria can help to accelerate protein function studies and antimicrobial research. We describe a new method for degrading bacterial proteins using plasmid-encoded degrader peptides which deliver target proteins for degradation by a highly conserved ClpXP protease. We demonstrate the mode of action of the degraders on a challenging essential target, GroEL. The studies in bacteria are complemented by in vitro binding and structural studies. Expression of degrader peptides results in a temperature-dependent growth inhibition and depletion of GroEL levels over time. The reduction of GroEL levels is accompanied by dramatic proteome alterations. The presented method offers a new alternative approach for regulating protein levels in bacteria without genomic modifications or tag fusions. Our studies demonstrate that ClpXP is an attractive protease for the future use in bacterial-targeted protein degradation.

**Keywords** Targeted Protein Degradation; Degrader; GroEL; ClpXP; Proteolysis
**Subject Categories** Methods & Resources; Microbiology, Virology & Host Pathogen Interaction; Post-translational Modifications & Proteolysis

## Introduction

Eliminating the protein of interest by inducing its degradation is the state-of-the-art approach for studying protein function. While CRISPR technology has revolutionised and facilitated gene knock-outs, targeted protein degradation (TPD) tools offer the ultimate progress for interrogating a biological problem: time- and dose-dependent control of protein levels. TPD is also a fast-evolving area in drug development, in which degrader molecules are used to destroy the target protein involved in pathogenesis. In bacteria, most approaches to precise, tunable, and inducible degradation rely on a fusion of the protein of interest with a degron (a degradation signal, often a short sequence motif) (Varshavsky, 1991; Izert et al, 2021). Several degron-based systems have been developed and used to study protein function, or in advanced synthetic biological circuits (Wei et al, 2011; Potvin-Trottier et al, 2016; Durante-Rodríguez et al, 2018; Lei et al, 2022). However, these approaches are only applicable to targets that can be safely tagged without affecting their function and require genetic modifications which could potentially introduce a bias in the experimental system. Similarly, the use of temperature-sensitive mutants requires genetic modification of the target, and may not be feasible if appropriate mutations cannot be found, or if the studied growth conditions are a limiting factor. Excitingly, the antibiotic pyrazinamide has been found to act by exposing a native degron in the target protein PanD upon binding of pyrazinoic acid, leading to PanD destabilisation and degradation (Gopal et al, 2020). However, it is difficult to design such destabilising compounds, so that small-molecule-induced instability is not yet a universal mechanism which could be extended to a wider range of bacterial targets. Thus, there is still an unaddressed demand for methods and molecular tools that allow the degradation of native, unmodified bacterial proteins.

In eukaryotic cells, a variety of molecular tools are available for TPD of selected proteins (Liu and Ciulli, 2023). These tools include bivalent chimeric molecules such as proteolysis-targeting chimeras (PROTACs) (Sakamoto et al, 2001), autophagy-targeting chimeras (Takahashi et al, 2019), and antibody-based degraders (Banik et al, 2020; Cotton et al, 2021), as well as monovalent molecular glues (Mayor-Ruiz et al, 2020; Słabicki et al, 2020). These so-called degraders have been extensively characterised and they are now entering the clinic (Chirnomas et al, 2023). Degraders are attractive drug candidates due to several advantages over conventional inhibitors, which rely on occupying the active site of the protein. Degraders allow the targeting of proteins previously considered as undruggable (such as transcription factors (Samarasinghe et al, 2022) or scaffolding proteins (Wang et al, 2019; Kim et al, 2023)). Degraders have also been reported to be effective already at low, sub-stoichiometric concentrations due to their distinct event-driven mode of action (Bondeson et al, 2015). Importantly, they act fast enough to decrease the emergence of drug resistance, which has been shown for both cancer and viral targets (de Wispelaere et al, 2019; Hughes et al, 2021).

[1]Structural Biology Group, Biological and Chemical Research Centre, Faculty of Chemistry, University of Warsaw, Warsaw, Poland. [2]Biological and Chemical Research Centre, Faculty of Chemistry, University of Warsaw, Warsaw, Poland. [3]Cryomicroscopy and Electron Diffraction Core Facility, Centre of New Technologies, University of Warsaw, Warsaw, Poland. [4]IMol Polish Academy of Sciences, Warsaw, Poland. ✉E-mail: mw.gorna@uw.edu.pl

In many respects, therefore, TPD is a promising approach for creating new antibiotics that would act through novel mechanisms and with improved properties. However, the conventional TPD approaches exploit protein degradation mechanisms specific to eukaryotic cells (such as ubiquitin-dependent proteasomal degradation or autophagy) and therefore cannot be easily adapted to act in bacteria. Alternatively, direct tethering of proteins to the proteasome has also been shown to be an effective strategy for proximity-induced degradation (Bashore et al, 2022). We previously hypothesised that TPD in bacteria could similarly be achieved by direct recruitment of a bacterial protease to the target (Izert et al, 2021). Indeed, the first bacteria-specific degraders, BacPROTACs, have recently been successfully used to target fusion proteins by tethering them to the mycobacterial protease ClpC1P (Morreale et al, 2022). These BacPROTACs incorporate the Cyclomarin A ligand specific for ClpC1, a protease component that does not exist in other bacterial phyla beyond Actinobacteria. However, this example provides a premise for the development of analogous methods for targeted degradation of endogenous bacterial proteins, which could be extended to other bacterial species.

Here, we report a novel PROTAC-like approach for the degradation of endogenous untagged proteins in *Escherichia coli* using plasmid-encoded Clp-Interacting Peptidic Protein Erasers (CLIPPERs) that directly engage the ubiquitous bacterial ClpXP protease. We demonstrate the effectiveness of our method on the example of an essential bacterial chaperone protein GroEL. Our method offers a straightforward approach for functional studies enabled by selective protein depletion in bacteria. CLIPPERs represent a starting point for a new generation of universal bacterial degraders and putative degrader antibiotics that act through the ClpXP pathway.

## Results and discussion

### Establishing a system for peptide-mediated TPD in *Escherichia coli*

In search of a universal bacterial TPD system, we embarked first on identifying effective methods to exploit endogenous proteolytic systems. Since bacteria do not possess the canonical ubiquitin-proteasome pathway (and the analogous Pup-proteasome pathway is restricted to Actinobacteria), it is necessary to explore other proteases as the recruited degrading machinery. To establish a TPD system in *E. coli*, we have selected a robust protease from the AAA+ family, ClpXP. ClpXP can degrade a number of different substrates (Flynn et al, 2003), but it is best known for its involvement in the ribosome-rescue tmRNA (SsrA) pathway, degrading ssrA-tagged polypeptides from stalled ribosomes (Gottesman et al, 1998). The ClpXP system and its interactions with ssrA-tagged substrates and the substrate-delivering adaptor SspB are highly conserved, and they have been effectively used for synthetic degron-based systems in a variety of species, both Gram-positive and Gram-negative (Kim et al, 2011; Griffith and Grossman, 2008; Davis et al, 2009, 2011). In the absence of any known suitable small-molecule ClpX ligands, we used peptides interacting with ClpXP to create a new class of degraders: Clp-interacting peptidic protein erasers (CLIPPERs). CLIPPERs were

designed by fusing three different components: (i) an anchor that interacts with the proteolytic machinery, (ii) a flexible linker, and (iii) a bait that binds to the target protein (Fig. 1A). To determine the optimal protease recruitment strategy, we screened anchors based on known peptides capable of interacting with either the SspB adaptor (the AANDENY fragment of the ssrA degron (Song and Eck, 2003), doubled in the AANDENYAANDENY peptide (Klimecka et al, 2021)), the ClpX protease (XB and its shorter fragment sXB (Park et al, 2007; Dougan et al, 2003)) or directly with the ClpP protease (fragments containing the IGF loop of ClpX and the IGL loop of ClpA (Kim et al, 2001)) (Fig. 1B, C). The anchors were cloned into the arabinose-inducible expression plasmids at the C-terminus of the eGFP reporter protein. Expression of such fusion constructs in *E. coli* followed by translation arrest resulted in a gradual loss of fluorescence of the eGFP (Fig. 1C). This allowed us to validate the potential anchors interacting with the proteolytic systems that could be exploited in TPD approaches. Among the anchors tested, the ClpX-binding peptide derived from the C-terminal fragment of the SspB adaptor (XB) was shown to induce a moderate but steady degradation of fused eGFP. This effect was suppressed in the $clpX^-$ and $clpP^-$ mutant *E. coli* strains (Fig. EV1A,B). We did not observe any effect of SspB deletion on XB efficiency (Fig. EV1C). XB-induced degradation was also suppressed in the presence of the protease inhibitor bortezomib (Fig. EV1D-F), confirming that the loss of fluorescence was due to proteolysis of the fusion construct. Although the eGFP-ssrA fusion resulted in the most efficient degradation, the ssrA peptide is directly engaged in the ClpX entry pore, so that ssrA-based CLIPPERs themselves would be highly susceptible to degradation. Moreover, even if such "suicidal" ssrA-based CLIPPERs were efficient, the location of the ssrA degron must be C-terminal for recognition by ClpX, which would limit the possible peptide order in CLIPPER design to the use of N-terminal baits only. In contrast, the XB peptide is engaged by the dimeric Zinc Binding Domain (ZBD) of ClpX regardless of the position of the XB motif in the polypeptide, and could reduce CLIPPER proteolysis while still inducing efficient proximity-based degradation of the target. The XB peptide was therefore selected as the leading candidate for use as a CLIPPER anchor in our peptide-based TPD.

### Peptide-mediated GroEL depletion dysregulates bacterial proteostasis

To test whether CLIPPERs containing the XB fragment could be used to deplete endogenous proteins in bacteria, we designed fusion peptides consisting of an N-terminal Myc peptide tag, followed by the XB anchor, a flexible linker and a bait peptide. Although peptides can potentially be used as antimicrobials, their effective import into bacteria can present challenges that have yet to be addressed (Lee et al, 2021; Oikawa et al, 2018). To overcome such limitations, we have created a series of arabinose-inducible expression plasmids encoding the tested fusion peptides to study the effect of CLIPPER expression on bacterial phenotypes. To demonstrate the mode of action of the peptide degraders, we chose the GroEL chaperone as a potential TPD target. GroEL is conditionally essential in *E. coli*, allowing a facile phenotypic readout of the GroEL protein loss in cells. GroEL is known to assist the folding process of multiple proteins, many of which are strictly

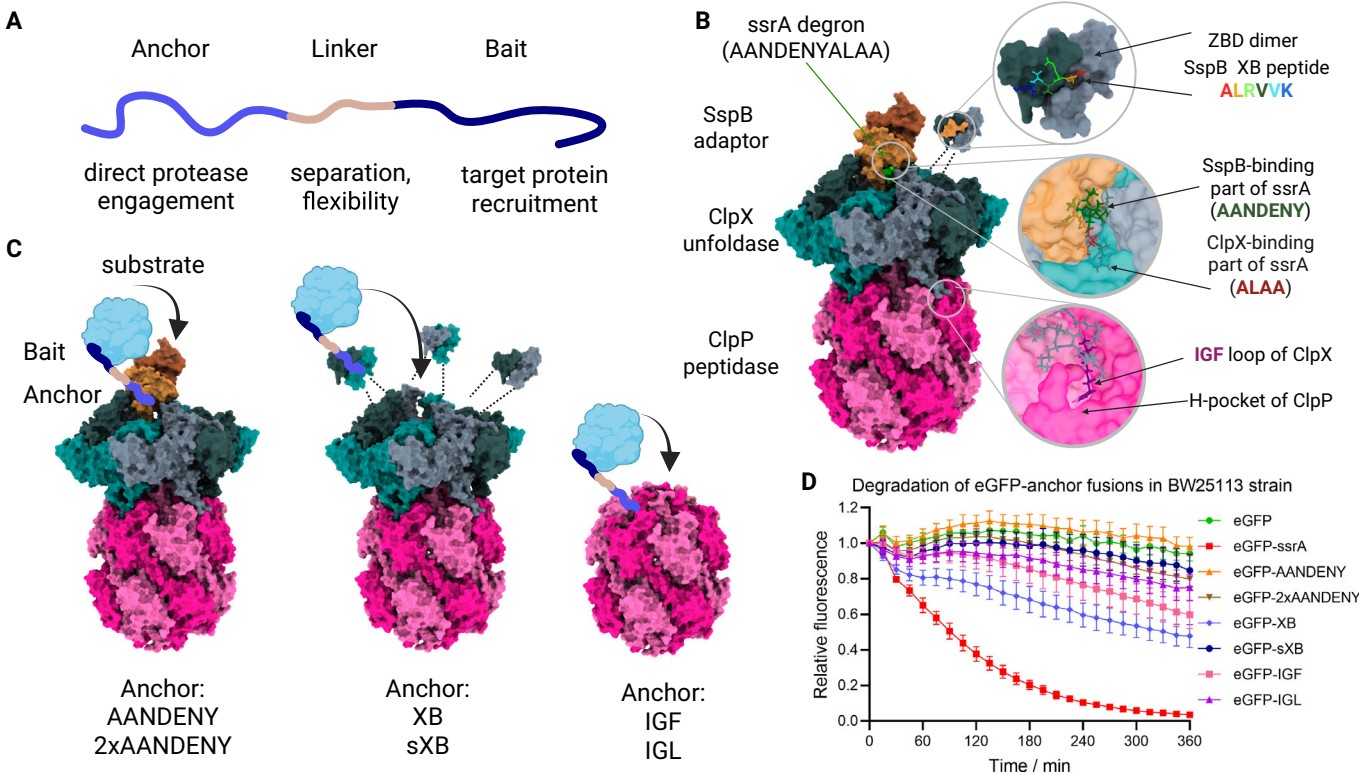

**Figure 1. The design of ClpXP-engaging peptidic protein degraders (CLIPPERs).**

(A) Schematic representation of the degrader peptides (CLIPPERs). The fusion peptide is composed of an "anchor" which recruits the protease, a "bait" which binds to the target protein and a linker moiety which separates the two peptides. (B) Representation of the ClpXP-SspB complex (based on the ClpXP-SspB-ssrA structure; PDB ID: 8ET3 (Ghanbarpour et al, 2023), modified). Indicated are the binding sites of the tested anchor peptides: the SspB-ClpX interface binding the ssrA degron, the Zinc Binding Domain of ClpX with the SspB ClpX-binding peptide (XB), and the IGF loop of ClpX bound in the hydrophobic pocket of ClpP (H-pocket). The C-terminal fragment (XB) of the SspB and ClpX Zinc Binding Domain (ZBD) were not visualised in the experimental structural model and a schematic representation was added based on the ZBD-XB crystal structure (PDB ID: 2DS8 (Park et al, 2007)). (C) Schematic representation of possible strategies for direct ClpXP recruitment by CLIPPERs binding to SspB, ClpX ZBD, or ClpP. (D) Degradation of the eGFP-anchor fusion proteins in bacteria. The protein levels were monitored by recording bacterial fluorescence following a translation arrest. The values at time point 0 were normalised to the initial sample fluorescence. The curves represent mean values from three independent biological repeats (averaged for clarity) with error bars representing SEM. Source data are available online for this figure.

dependent on the GroEL (Kerner et al, 2005; Fujiwara et al, 2010; Niwa et al, 2016). GroEL has been identified as a potential antimicrobial target, although no GroEL-targeting therapeutics are currently in use (Kunkle et al, 2018; Abdeen et al, 2016). Targeting an essential protein that regulates multiple other proteins appeared to be an attractive strategy to observe the phenotypic changes even with moderate degradation of the target. In addition, GroEL has a well-defined peptide ligand, the "strong binding peptide" (SBP) (Chen and Sigler, 1999), which binds to GroEL without disrupting its activity. Our preliminary experiments for degraders of another *E. coli* chaperone, DnaK, using its known peptide ligands (Gragerov et al, 1994; Otvos et al, 2000), have shown that using highly toxic baits as part of degrader constructs can make it difficult to distinguish the effect of the bait from the degradation event (Appendix Fig. S1A, B). Therefore, targeting GroEL with the relatively non-toxic SBP peptide appeared to be a suitable approach to demonstrate the action of CLIPPERs.

To test whether GroEL-targeting CLIPPERs (GroTAC1 and GroTAC2, including flexible GGS and GGSGGSGG linkers, respectively; Fig. 2A) can affect bacterial growth, we have assessed *E. coli* growth in the presence or absence of the expression-inducing

arabinose. The initial screen showed that expression of GroTACs can impair the growth of bacterial colonies in a temperature-dependent manner (Fig. 2B). We did not observe any effect of either the XB anchor or the SBP bait on bacterial growth. The initial results were confirmed by experiments in liquid cultures. Expression of GroTACs with two different linker lengths resulted in a moderate growth inhibition reaching around 20–25% when bacteria were cultured at a permissive temperature of 30 °C (Fig. 2C). This effect was not observed in the *clpP*- mutant strain (Fig. EV2). The observed growth reduction was potentiated when bacteria were cultured at an elevated temperature of 42 °C, when proteins are more prone to misfolding and the folding activity of GroEL is more essential (Fig. 2D). The observed growth reduction depended on the level of peptide expression (Fig. 2E), although the regulation of pBAD vectors does not show a linear response to inducer concentration (Guzman et al, 1995). The changes in bacterial growth observed by monitoring cell culture density were confirmed by an enzymatic viability assay (Fig. 2F). Expression of the degraders at 42 °C caused a significant reduction in *E. coli* viability, even greater than the reduction in culture density. This allowed us to conclude that GroTACs affect bacterial fitness under

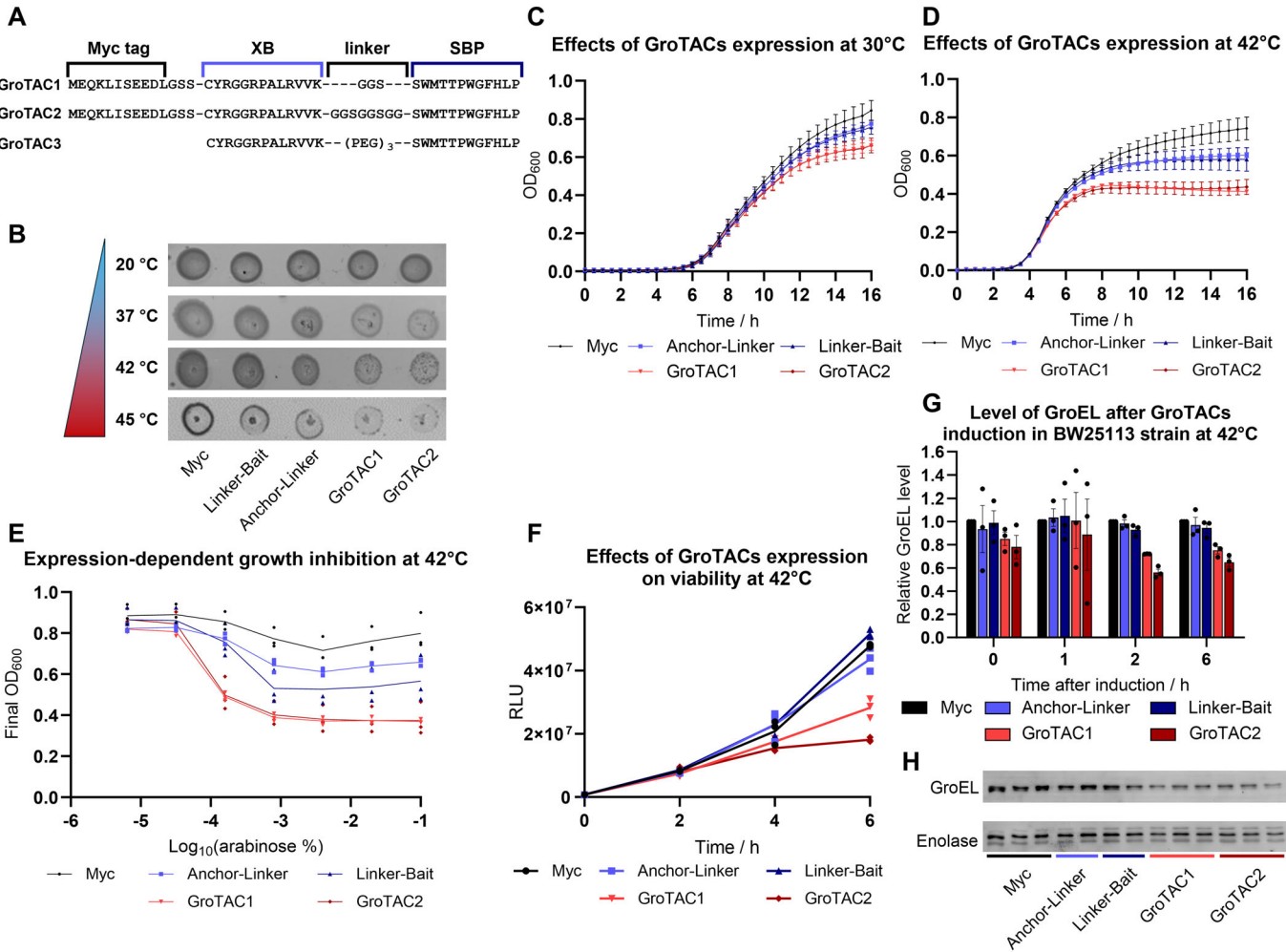

**Figure 2. *E. coli* BW25113 bearing plasmids with tested peptides (Myc control peptide, Anchor-Linker, Linker-Bait and GroTACs with two linker lengths) were subjected to growth tests and GroEL levels measurement.**

(A) A schematic of the composition of the GroTAC peptides. (B) Drop test of bacteria grown on LB plates with 0.02% arabinose incubated at different temperatures. (C) Growth curves of bacteria grown at 30 °C in the presence of 0.02% arabinose. The curves represent mean values from three independent biological repeats (averaged for clarity) with error bars representing SEM. (D) Growth curves of bacteria grown at 42 °C in the presence of 0.02% arabinose. The curves represent mean values from three independent biological repeats (averaged for clarity) with error bars representing SEM. (E) The effect of arabinose concentration on bacterial growth at 42 °C. The curves represent mean values from three independent biological repeats. (F) BacTiter-Glo enzymatic viability test of bacteria at different time points after induction of peptide expression at 42 °C. The curves represent mean values from three independent biological repeats. (G) Western blot measurement of GroEL level relative to total protein at different time points after induction of peptide expression. Quantified results were normalised to the level of GroEL in bacteria expressing Myc peptide. The bars represent mean values of three independent biological replicates, and error bars represent SEM. (H) A representative western blot of GroEL levels in total protein extracts from bacteria expressing GroTACs or control peptides at 42 °C, 6 h from induction, in technical replicates. Source data are available online for this figure.

thermal stress. To test whether the observed differences were caused by changes in GroEL levels, we performed western blot analysis at several different time points after the induction of bacteria cultured at 42 °C. We observed a progressive decrease in GroEL protein levels in bacteria expressing GroTACs compared to bacteria expressing the control peptides (Figs. 2G, H and EV3). The reduction reached up to 40% of the GroEL level in comparison to the control groups, which was consistent with the viability experiments. In addition, visualisation of the whole protein content on the membranes by a stain-free technique revealed dramatic alterations in the protein pattern in GroTAC-expressing bacteria (Fig. EV3). The alterations and apparent loss of proteins were

observed regardless of the normalisation strategy used (normalisation to protein content or to bacterial cell number). This suggests that even a modest loss of GroEL can have a dramatic effect on the bacterial proteome.

To verify how the GroEL depletion affected the levels of other proteins, we have performed a quantitative shotgun proteomics analysis of the bacterial proteomes at 1 h (Fig. 3A, D), 2 h (Fig. 3B, E) and 6 h (Fig. 3C, F) after induction of peptide expression in *E. coli* grown at 42 °C. We identified 2357 protein groups matched to the reference proteome, including 1857 that were consistently quantified across all replicates and experimental conditions. We also identified a tryptic peptide LISEEDLGSSCYR

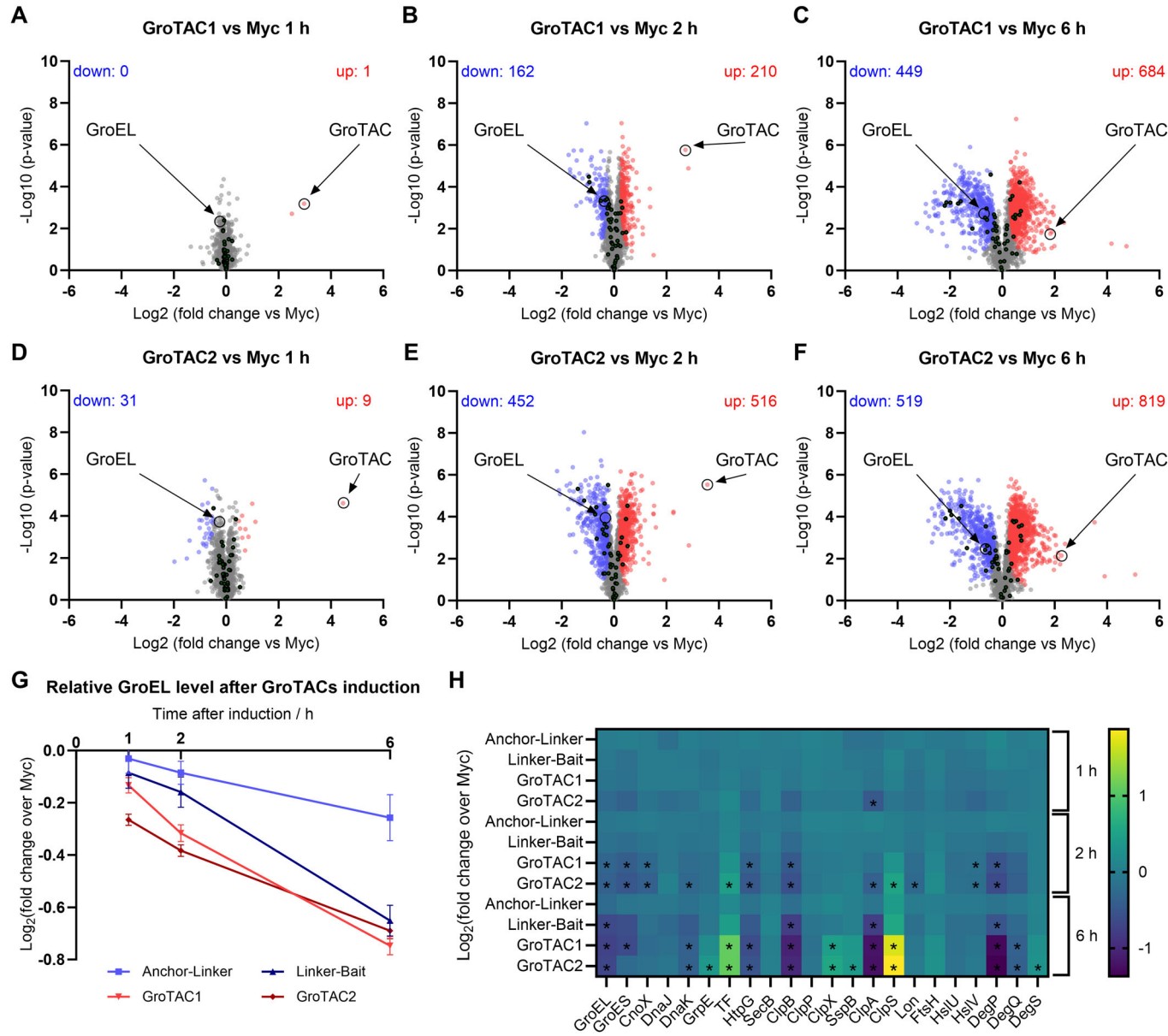

**Figure 3. Proteomics analysis of *E. coli* BW25113 expressing GroTACs.**

(A–F) Volcano plots displaying the log2 fold change (log2FC, *x* axis) against the *t* test-derived −log10 statistical *P* value (*y* axis) for protein groups detected in total lysates of *E. coli* expressing GroTAC1 or GroTAC2 for 1 h ((A), (D), respectively), 2 h ((B), (E), respectively), and 6 h ((C), (F), respectively) compared to *E. coli* expressing control peptide, Myc, by TMT-MS analysis (*n* = 3). Student's *t* test (two-sided, unpaired) was performed for the statistical analysis. Protein levels found upregulated, downregulated (*q* value < 0.05), and unchanged are indicated as red, blue, and grey dots, respectively. The obligate GroEL substrates are marked with black dots. GroEL and GroTAC are marked with open circles. (G) Relative changes in GroEL levels in bacteria expressing GroTACs and control peptides. Values from the TMT-MS biological triplicates (*n* = 3) were averaged for clarity, and the error bars represent SEM. (H) Heatmap representation of changes in levels of proteins involved in cellular proteostasis, statistically significant changes (*q* value < 0.05, tested as described above) are marked with asterisks. Source data are available online for this figure.

derived from the XB motif of the GroTACs, providing additional evidence for their reproducible expression under the chosen experimental conditions. Analysis revealed that GroTAC expression led to a progressive dysregulation of the bacterial proteome with hundreds of proteins up- or downregulated, which was not observed in the bacteria expressing the control peptides (Appendix Fig. S2). After a 6 h expression of the Linker-Bait peptide, we observed a decrease in the levels of several proteins, suggesting that

the Anchor peptide itself could potentially interfere with the cellular functions of GroEL. The MS results confirmed the progressive GroTAC-induced decrease in GroEL levels observed in western blot experiments (Fig. 3G). Among the altered proteins, besides GroEL, we observed a progressive downregulation of the obligate GroEL substrates, known as class IV substrates (Fujiwara et al, 2010) (Figs. 3A–F and EV4). Other classes of substrates can either fold spontaneously or be assisted by other bacterial

chaperones (Kerner et al, 2005; Fujiwara et al, 2010), thus we did not observe significant overall changes in their relative levels. The downregulation of obligate substrates was not observed at 1 h but appeared at 2 h and was even more pronounced at 6 h after induction, which may point to a secondary effect of GroEL depletion that takes time to develop. Thus, even partial GroEL depletion can affect the levels of its clients, presumably by impairing their folding, which can lead to aggregation or proteolytic degradation. The downregulation of GroEL was accompanied by changes in the levels of a number of other proteins involved in the regulation of cellular proteostasis (Fig. 3H). We observed that the GroEL-interacting co-chaperonins GroES and CnoX (YbbN) were also downregulated in GroTAC-expressing bacteria. The reduction in their levels could be an additional effect of the induced proximity between GroEL and ClpXP, which could in principle facilitate the degradation of target-bound proteins. Several other chaperones were altered, including downregulated DnaK, GrpE, HptG and ClpB. At the same time, another chaperone, trigger factor (TF), was significantly upregulated together with its substrates, ribosomal proteins (Martinez-Hackert and Hendrickson, 2009) (Fig. EV4). This could indicate the activation of a rescue mechanism to compensate for the loss of GroEL. We also observed changes in the levels of several proteins involved in proteolysis such as Lon, DegP, and ClpP with its partners ClpA, ClpX, and ClpS. This suggests that GroTAC-mediated depletion of GroEL can induce massive changes in cell proteostasis, leading to severe growth dysfunctions and cell death under thermal stress. Since GroEL interacts with hundreds of protein partners involved in a wide variety of processes, even incomplete depletion of GroEL had severe effects on the bacterial proteostasis.

## In vitro validation of GroTAC action suggests the mechanistic limitations of CLIPPERs

To validate the mode of action of GroTACs, we performed a series of in vitro experiments using purified peptides and proteins. The first prerequisite for GroTAC action is binding to its protein partners ClpX and GroEL. We assessed the binary interactions of His-SUMO-tagged GroTACs or their components with their respective partners using biolayer interferometry (BLI) (Fig. 4; Appendix Table S1). Isolated ClpX bound to the His-SUMO-XB-GGS (Anchor-Linker) peptide with $K_D$ of ~65 nM (Fig. 4A). GroTAC1 (His-SUMO-XB-GGS-SBP) had a similar affinity for ClpX ($K_D$ ~70 nM) (Fig. 4B), suggesting that the XB motif retains its binding capacity even in fusion with other peptides. Similar affinities were obtained for binding of untagged, synthetic GroTAC1 ($K_D$ of 35 nM), GroTAC2 (53 nM), and GroTAC3 (82 nM) peptides to immobilised His-tagged ZBD (Fig. EV5A–C; Appendix Table S1), further confirming the specificity of interactions between the peptides and ClpX and that an N-terminal tag present on a GroTAC does not affect its binding to ClpX. The binding event alone may not be sufficient to trigger the ClpXP-mediated degradation of tethered target proteins. The SspB adaptor is known to activate the ClpXP complex by stimulating its ATPase activity, and the XB peptide alone can also activate ClpX. We have tested if His-SUMO-GroTAC1 retains similar ClpX-activating properties. To our surprise, GroTAC1 showed an even stronger stimulation of ClpX ATPase activity than the Anchor-Linker (His-

SUMO-XB-GGS) peptide alone, although the Linker-Bait (His-SUMO-GGS-SBP) peptide had no effect on ClpX (Fig. 4C). Thus, GroTAC1 not only binds but also activates ClpX, increasing its potential usefulness as a degrader. GroEL was bound by Linker-Bait (His-SUMO-GGS-SBP) with a $K_D$ of 94 nM (Fig. 4D; Appendix Table S1). GroTAC1 had a slightly reduced affinity towards GroEL (154 nM), but the $K_D$ was still in a similar range (Fig. 4E). At the same time, we observed a significant deviation from the 1:1 binding stoichiometry, which could indicate uneven occupancy of the SBP-binding sites of the GroEL subunits, but could also be due to steric constraints (only one side of the GroEL barrel can face the peptide-covered sensor surface). This led us to conclude that fusion of the XB anchor with the SBP bait did not significantly perturb the interactions of these peptide fragments with their respective protein partners and that GroTAC1 retained similar binding capacities as the individual peptide ligands. Next, we measured the formation of the ternary complex between GroEL-bound GroTAC1 or GroTAC2 and immobilised ZBD. We observed a higher BLI signal during association with ZBD for the GroEL-GroTAC complexes compared to the peptides alone, indicating the GroTAC-induced formation of the ternary complex (Fig. EV5D), and the ternary affinities were 165 nM and 308 nM for GroTAC1 and GroTAC2, respectively. This suggests that GroTACs can simultaneously bind to both ClpX and GroEL and induce their interactions, thereby driving GroEL degradation in cells. Finally, we attempted to reconstitute the protease system to verify if we could observe degradation in vitro using purified peptides and proteins: ClpXP complex, GroTAC1 and GroEL. Under the conditions tested, we observed only a modest degradation of GroEL, reaching ~40% after a 10 h incubation in a reaction mixture (Fig. 4F).

To better understand the binding topology between GroTAC and GroEL, we obtained a structural model of their complex using cryogenic electron microscopy. The resulting electrostatic potential map revealed an additional volume at the top of the GroEL barrel lumen, in the region of the GroEL apical domain. This was consistent with the binding site of the SBP peptide in the co-crystal structure of GroEL in complex with SBP (PDB ID 1MNF) (Fig. 4H; Appendix Fig. S3). The low resolution of our experimental map in this region might be due to reduced occupancy and/or flexibility of the GroTAC3 molecule. The additional density attributed to the ligand corresponded to the size and orientation of SBP in the available co-crystal structure. Ab initio modelling tools such as AlphaFold or ModelAngelo consistently indicated the presence of a proline in the position corresponding to the middle of the SBP peptide in the crystal structure (Fig. 4H). Based on the tracing of the peptide backbone and the positioning of P22 in the middle of SBP, we were able to model the SBP part of GroTAC3 in the corresponding electrostatic potential density. The results clearly confirmed that GroTAC can effectively bind to the GroEL target. However, the XB-linker segment of GroTAC could not be visualised due to extremely low resolution—suggesting the expected high flexibility and mobility of the unbound remainder of GroTAC in the absence of ClpX.

Modification of protein stability by specific peptidic degrons, such as ssrA, is a well-established strategy used to study protein function (Izert et al, 2021). A common inducible degradation system relies on constitutive expression of an ssrA-tagged protein of interest and inducible expression of the SspB adaptor (McGinness et al, 2006). However, studying protein function in

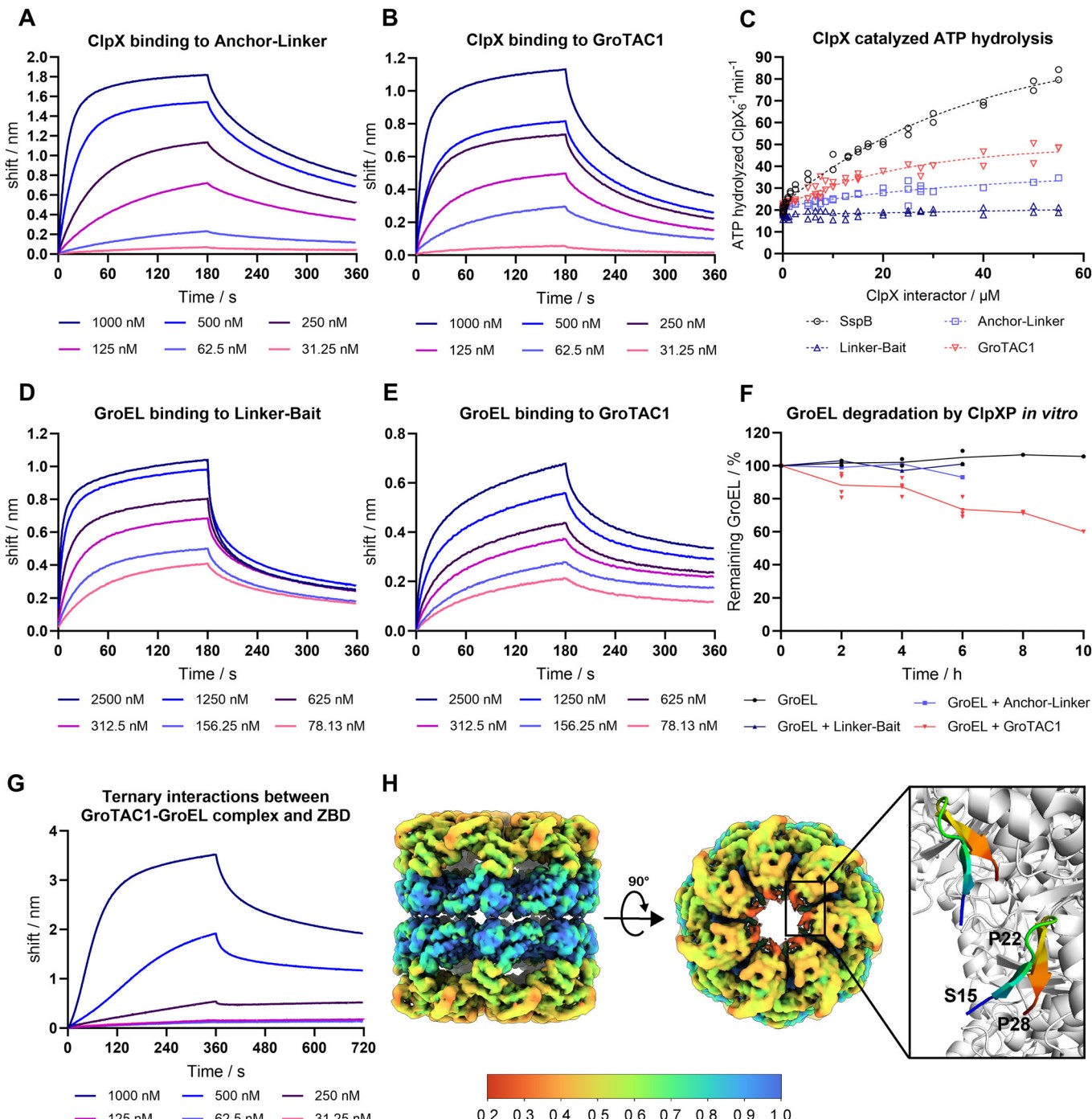

the *sspB-* genetic background, when protein turnover can be reduced, may introduce some bias. In addition, this approach requires genetic modifications (i.e. fusions) of target proteins in bacteria. Introducing tags can alter protein folding, activity, or localisation (Swulius and Jensen, 2012; Weill et al, 2019; Singh et al, 2020). For example, in our hands, purified GroEL fused to a purification tag could not be assembled into functional barrels. Here, we propose a novel method for inducible, peptide-mediated depletion of native, unmodified, endogenous bacterial proteins.

We used the SspB-derived "XB" peptide which binds and activates the ClpX component of the bacterial ClpXP proteolytic complex. The interactions between SspB and ClpX are highly conserved, so that heterologously expressed SspB can interact with ClpX from bacteria that normally lack this adaptor (Kim et al, 2011; Griffith and Grossman, 2008). This provides a premise for the broad universality of the proposed approach. The XB peptide has previously been used as an FKBP12-XB fusion for rapamycin-induced degradation of ssrA-tagged proteins using a split SspB adaptor system (Davis et al, 2011). This supported our further

**Figure 4.  In vitro analysis of GroTACs interactions with ClpX and GroEL.**

(A) Sensograms representing the BLI measurement of ClpX binding to His-SUMO-XB-GGS (Anchor-Linker). (B) Sensograms representing the BLI measurement of ClpX binding to His-SUMO-XB-GGS-SBP (GroTAC1). (C) Stimulation of ATPase activity by a natural ClpX partner (SspB), His-SUMO-XB-GGS-SBP (GroTAC1), His-SUMO-XB-GGS (Anchor-Linker), or the negative control His-SUMO-GGS-SBP (Linker-Bait). The points represent the average values from technical triplicates, and the dotted lines connect mean values from the biological replicates. (D) Sensograms representing the BLI measurement of GroEL binding to His-SUMO-GGS-SBP (Linker-Bait). (E) Sensograms representing the BLI measurement of GroEL binding to His-SUMO-XB-GSS-SBP (GroTAC1). (F) In vitro degradation of GroEL by ClpXP in the presence of His-SUMO-XB-GGS-SBP (GroTAC1), His-SUMO-XB-GGS (Anchor-Linker), or His-SUMO-GGS-SBP (Linker-Bait). The points represent individual values from two independent replicates (4 replicates in the case of GroTAC1 up to 6 h). The lines connect mean values (averages from the independent replicates). (G) Sensograms representing the BLI measurement of complex formation between GroEL and immobilised ZBD in the presence of increasing concentrations of the untagged, synthetic GroTAC1 peptide. (H) Electrostatic potential density map obtained by Cryo-EM, coloured by relative occupancy, and visualisation of GroTAC3 molecules (rainbow) binding to GroEL subunits based on the obtained structure from Cryo-EM map. Relative occupancy was estimated using OccuPy (Forsberg et al, 2023). Source data are available online for this figure.

premise that induced proximity by tethering to ClpXP would be sufficient for engineering bacterial TPD, although a degron-less target has not been attempted previously. In our study, we used the XB peptide in CLIPPERs in direct fusion with a target-interacting peptide to cause degradation of unmodified endogenous proteins. This shows that tethering the native target protein to the activated ClpXP protease can induce target degradation. Such an effect has already been demonstrated in the recent pioneering work on bacterial small-molecule degraders (BacPROTACs) exploiting the ClpC1P protease in Mycobacteria and ClpCP in Gram-positive bacteria (Morreale et al, 2022; Hoi et al, 2023). BacPROTACs are an exciting new type of putative antibiotics based on a completely different mechanism of action to classical inhibitors. Given also the success of pyrazinamide as a commonly used antibiotic against TB (Gopal et al, 2020), broadening the spectrum of the TPD repertoire against more bacterial orders is of critical importance for future antibiotic development. The most lethal antimicrobial-resistant pathogens are predominantly Gram-negative species (Ikuta et al, 2022) that do not express ClpCP, and therefore cannot be targeted by the currently known BacPROTACs. Our work opens the door to TPD based on a Clp family protease also in Gram-negative bacteria, and suggests that the ClpX-XB interactions may be worth exploring for the design of specific ClpX-binding small molecules for potential use in new BacPROTACs.

We demonstrated the use of our new method on an essential bacterial chaperone, GroEL, which is a potential antimicrobial target (Abdeen et al, 2016; Kunkle et al, 2018). Using a well-characterised peptidic ligand of GroEL (SBP) as the bait in an XB-based CLIPPER resulted in progressive loss of GroEL upon CLIPPER expression. This approach was effective as early as 1 h post-induction against a high-copy target such as GroEL, which is present in cells at more than an order of magnitude higher protein levels than ClpXP itself (Lorimer, 1996; Farrell et al, 2005), while the concentration of GroEL is further increased up to tenfold at high temperatures. A model GFP-ssrA substrate has been reported to be degraded by the cellular ClpXP pool at a rate of 100,000 molecules per generation in *E. coli* cells doubling every 20 min (Farrell et al, 2005). In our hands, GroTAC2 expression reduced GroEL protein levels by ~17% in 1 h in cells growing at 42 °C. Even allowing for the rapid synthesis of new GroEL copies (~50,000–60,000 molecules per cell generation (Li et al, 2014)), which may mask the true in vivo degradation rate, this result still suggests that a native degron-less protein is indeed a markedly more difficult target than the typical ssrA-tagged substrate (in our hands, fluorescence of overexpressed eGFP-ssrA decreased by

~40% in 1 h; Fig. 1D). This difference could be due to a slower initiation of degradation, as unstructured terminal tags could more readily engage the protease pore. However, a moderate degradation efficiency that removes only a fraction of the target protein is not atypical of early PROTAC prototypes, as seen in many TPD studies in human cells. In bacteria, the first BacPROTACs (Morreale et al, 2022) have achieved at most ~60% reduction of the target fusion protein at their maximum concentration (100 μM compound). Quantitative MS analysis of the only working BacPROTAC-3 prototype has shown less than a twofold reduction of the fusion target. In the follow-up study (Hoi et al, 2023), Homo-BacPROTACs have only been able to reduce the levels of ClpC1 and ClpC2 up to a similar range—40% and 45–60%, respectively—after a 24 h treatment. The GroTAC example suggests that CLIPPERs may act with a similar apparent efficacy as the small-molecule BacPROTACs, of which there are only a few working examples to date. Going beyond these proof-of-concept studies, future TPD work in bacteria may turn to increasing the potency of bacterial degraders, but this may require extensive optimisation, as is the case for many TPD agents in human cells. Nevertheless, the first demonstration of CLIPPERs has also shown their antimicrobial potential. Even moderate GroEL depletion resulted in reduced bacterial survival and dysregulation of the *E. coli* proteome. This is consistent with the GroEL function as a central hub in maintaining bacterial proteostasis in environmental stress. CLIPPER-mediated protein downregulation can therefore be used to study the function of proteins of interest or to validate antibiotic targets, advantageously in a time-resolved manner and in the case of essential proteins or those not amenable to tagging. As the CLIPPER design was relatively straightforward and modular, we expect it to be easily extended to other targets using their known interacting peptides as baits.

Our study also reveals some of the limitations and requirements of using TPD in bacteria and raises further questions. We hypothesise that the highly stable and well-ordered GroEL barrels may be naturally resistant to degradation by ClpXP. Rather, we suggest that the fraction of dissociated, monomeric GroEL would undergo GroTAC-mediated degradation, which could explain the limited degradation observed both in vitro and in bacteria. Higher temperatures destabilise the GroEL complexes (Walker et al, 2022) and stimulate the expression of GroEL (Martin et al, 1992). We suspect that in bacteria we observed predominantly GroTAC-mediated degradation of newly translated or unassembled GroEL subunits, which may be easier to engage and deliver to ClpXP (Fig. 5). A similar phenomenon has recently been shown for human

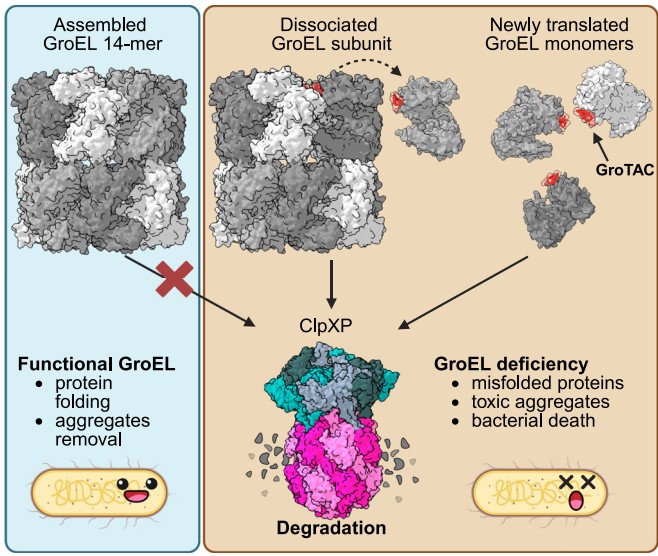

**Figure 5.** Schematic representation of GroTACs mode of action.

chaperones, which are resistant to degradation in the assembled state, but are recognised by their cognate adaptors and delivered for ubiquitination in the free monomeric state (Pla-Prats et al, 2023; Yagita et al, 2023). This may also suggest that, in addition to the half-life of the target protein, which is a known factor limiting TPD efficiency in human cells (i.e. depletion of short-lived targets has less of a marked effect (Schneider et al, 2021)), target synthesis rates may also be critical in bacteria, and these may vary more widely in prokaryotes in response to environmental stimuli. Differences in overall metabolic rates between bacterial species can also affect TPD applications. For example, *Mycobacterium tuberculosis* is known to divide slowly (every 18–24 h), and the effects of BacPROTACs are most pronounced and often analysed after 24 h of treatment (Morreale et al, 2022; Hoi et al, 2023). In our study, after 6 h of CLIPPER expression, the changes in *E. coli* viability and proteome composition have already progressed far beyond the primary effects of GroEL depletion, which we attribute to the much faster metabolism of this model bacterium. We expect the TPD efficacy to be both target- and species-dependent. In *E. coli*, the synthesis rate of GroEL is high under normal conditions (~52,000 copies per generation (Li et al, 2014)), which may have facilitated GroTAC action by promoting access to unassembled subunits. On the other hand, targets with a high protein copy number may also be more resistant to the TPD approach, depending on how well the target protein pool can buffer the demand for its function. However, due to fitness requirements, bacteria may have optimised the protein synthesis burden (Li et al, 2014), leaving little redundancy in some cases, as might have been the case with GroEL. Exploiting conditions that increase the demand for the target (such as elevated temperatures for GroEL) can help identify the most efficacious TPD applications.

Several other protein targets could not be degraded in our hands by other CLIPPER prototypes tested, suggesting that the range of proteins susceptible to CLIPPER-mediated degradation might be limited and/or their degradation may depend on specific culture conditions. A number of intrinsic factors can determine protein

stability and susceptibility to proteolysis. For instance, although ClpXP can extract ssrA-tagged proteins from membranes (Chai et al, 2016) or protein complexes (Moore et al, 2008), the efficacy of membrane protein degradation depends on protein topology and accessibility (Abeywansha et al, 2018). Similarly, the protein fold determines the degradation potency as ClpXP engages its substrates by either the N- or C-terminus (Flynn et al, 2003). Both the N- and C-terminus of GroEL face the interior of the GroEL barrel, making it difficult to engage without first disassembling the multimer. The small-molecule BacPROTACs have been shown to induce target degradation with varying potency depending on the substrate fold, with a preference for unstructured proteins (Morreale et al, 2022). Thus, the potency of degraders that directly engage proteases is not universal towards all targets, as some may be inherently more degradation-resistant. Moreover, the presentation of an unstructured region to the protease may also require a favourable stereochemistry of the ternary complex, aligning the target with the protease pore, and thereby facilitating initiation of degradation. Previously, we hypothesised that initiation of degradation may be the rate-limiting factor in protease-based TPD (Izert et al, 2021). Since the similar binary and ternary affinities of GroTAC1 and GroTAC2 do not explain why the latter is more efficient in mediating GroEL degradation in vivo, additional factors must play a role. For example, the longer linker in GroTAC2 may enable a better reach in bridging GroEL with ClpX, or presenting a more favourable orientation of the target. We propose that the plasmid-expressed CLIPPERs should be best suited for targeting less structured, unassembled, and cytoplasmic proteins, and that linker optimisation may bring rewarding results; however, all these concerns are typical for most TPD approaches. Finally, the identification of baits with sufficient binding affinity may also play a role in extending the application of CLIPPER technology to additional targets. This would require either prior knowledge of interacting peptides or the development of bait peptides using experimental methods such as phage display—as was done for SBP and GroEL. The advantage of a TPD approach is that more potential binding pockets, including neutral sites, can be explored in the target protein. CLIPPERs eliminate the need to target the active site, thus extending the target range to non-enzymes, including many cases classically considered "undruggable". However, the limitation to ligandable targets remains, as good and specific binding is still a prerequisite for TPD. Fortunately, peptides are thought to offer a relatively easier way to develop good binding properties, as they typically have larger interaction surfaces than small molecules. All of this may encourage researchers to try out CLIPPERs, first as a useful tool in biotechnology. Next, CLIPPERs may pave the way for the development of more drug-like TPD agents based on analogous small molecule chimeras, which may have a better chance of penetrating into the bacteria. However, it is worth noting that even the small-molecule BacPROTACs vary in their efficiency depending on the analogue used, as they exploit the active import of Cyclomarin A, and chemical modifications can alter their permeability (Junk et al, 2024). Finally, future research developments may also find ways to efficiently deliver peptides into the bacterial cytoplasm, which would overcome the ultimate limitation of CLIPPERs as titratable TPD tools and drugs. Possibly, a carrier molecule attractive to bacteria could be attached to a CLIPPER to facilitate active import in a "Trojan horse" approach. Alternatively, cell-penetrating peptides that can enter prokaryotic

cells have been developed (Lee et al, 2021; Oikawa et al, 2018), and such peptides could be fused to or envelop CLIPPERs to deliver CLIPPER cargo. However, such peptides often have toxic antimicrobial activity on their own, which could obfuscate the effect of CLIPPERs in our proof-of-concept study—as was the case with DnaK ligands. As drug delivery in bacteria is still an actively developing field, we leave these next tasks to future work on peptide BacPROTACs and focus on the TPD aspects of our new method.

In summary, we have achieved the first demonstration of a designed TPD-like approach acting on an unmodified target protein in a model Gram-negative bacterium. CLIPPERs represent a new way to control protein function through degradation, which may enable functional studies of bacterial proteins or may find applications in biotechnology and synthetic biology. Our method also provides a new tool for drug target validation and, in particular, could help to assess conditions and target suitability and degradability during the development of new bacterial degraders. The eventual replacement of peptides with small molecules would be a promising strategy for creating next-generation ClpXP-engaging bacterial degraders that could be applied to a variety of bacterial species, including the most threatening Gram-negative pathogens.

# Methods

### Reagents and tools table

| Reagent/resource | Reference or source | Identifier or catalogue number |
|---|---|---|
| **Experimental models** | | |
| *E. coli* Top10 | Laboratory strain collection | F⁻ mcrA DE(mrr-hsdRMS-mcrBC) φ80lacZ DE(M15) DE(lacX)74 recA1 araD139 DE(ara-leu)7697 galU galK λ⁻ rpsL(Str^R) endA1 nupG |
| *E. coli* BL21 (DE3) | Laboratory strain collection | F⁻ ompT hsdS_B (r_B⁻, m_B⁻) gal dcm (DE3) |
| *E. coli* C43 (DE3) | Laboratory strain collection | F⁻ ompT hsdS_B (r_B⁻, m_B⁻) gal dcm (DE3) *lac*UV5^p |
| *E. coli* BW25113 | Keio collection (Baba et al, 2006) | F⁻ rrnB DElacZ4787 HsdR514 DE(araBAD)567 DE(rhaBAD)568 rph-1 |
| *E. coli* J42W07 | Keio collection (Baba et al, 2006) | BW25113 DE(clpP)::kan |
| *E. coli* JW0428 | Keio collection (Baba et al, 2006) | BW25113 DE(clpX)::kan |
| *E. coli* JW0866 | Keio collection (Baba et al, 2006) | BW25113 DE(sspB)::kan |
| **Recombinant DNA** | | |
| pBAD-6xHis-SUMO-ClpX | Klimecka et al, 2021 | |
| pET28a-6xHis-SUMO-ClpP | Klimecka et al, 2021 | |
| pET28a-6xHis-SUMO-SspB | Klimecka et al, 2021 | |
| pET28a-His-ZBD | This study | |
| pET28a-GroEL | This study | |
| pET28a-6xHis-SUMO-XB-GGS | This study | |
| pET28a-6xHis-SUMO-GGS-SBP | This study | |
| pET28a-6xHis-SUMO-XB-GGS-SBP | This study | |
| pBAD-His-TEV-cAbGFP | This study | |
| pBAD-6xHIS-TEV | Klimecka et al, 2021 | |
| pBAD-6xHis-TEV-eGFP | Addgene | Cat #54762 |
| pBAD-6xHis-TEV-eGFP-ssrA | Klimecka et al, 2021 | |
| pBAD-6xHis-TEV-eGFP-AANDENY | Klimecka et al, 2021 | |
| pBAD-6xHis-TEV-eGFP-2xAANDENY | Klimecka et al, 2021 | |
| pBAD-6xHis-TEV-eGFP-XB | This study | |
| pBAD-6xHis-TEV-eGFP-sXB | This study | |
| pBAD-6xHis-TEV-eGFP-IGF | This study | |
| pBAD-6xHis-TEV-eGFP-IGL | This study | |
| pBAD-myc | This study | |
| pBAD-myc-XB-GGS | This study | |
| pBAD-myc-GGS-SBP | This study | |
| pBAD-myc-XB-GGS-SBP | This study | |
| pBAD-myc-XB-GGSGGSGG-SBP | This study | |
| pBAD-myc-NRLLLTG | This study | |
| pBAD-myc-XB-GGS-NRLLLTG | This study | |
| pBAD-myc-XB-GGSGG-NRLLLTG | This study | |
| pBAD-myc-pyrrhocoricin | This study | |
| pBAD-myc-XB-GGS-pyrrhocoricin | This study | |
| pBAD-myc-XB-GGSGGSGG-pyrrhocoricin | This study | |
| **Chemicals, enzymes and other reagents** | | |
| **Antibodies** | | |
| Mouse anti-GroEL | Invitrogen | MA3-022 |
| Rabbit anti-enolase | Gift from Professor Ben Luisi | |
| Goat anti-rabbit Alexa Fluor 488 | Invitrogen | A11008 |
| Goat anti-mouse Alexa Fluor 647 | Invitrogen | A32728 |

| Reagent/resource | Reference or source | Identifier or catalogue number |
|---|---|---|
| **Oligonucleotides** | | |
| PCR primers | Thermo Fisher Scientific; this study | Appendix Table S3 |
| Oligonucleotides | Thermo Fisher Scientific; this study | Appendix Table S3 |
| **Peptides** | | |
| GroTAC3 | KareBay Biochem; this study | Appendix Table S4 |
| Synthetic GroTAC1, GroTAC2, Anchor-Linker, Linker-Bait | GeneScript; this study | Appendix Table S4 |
| **Software** | | |
| Image Lab | BioRad | |
| Graphpad Prism 9 | Graphpad Software | |
| MaxQuant v. 1.6.17.0 | Tyanova et al, 2016a | |
| Perseus v.1.6.10.0 | Tyanova et al, 2016b | |
| Octet Analysis Studio | Sartorius | |
| cryoSPARC 4.4 | Structura Biotechnology Inc. | |
| OccuPy | Forsberg et al, 2023 | |
| ModelAngelo | Jamali et al, 2024 | |
| cg2all | Heo and Feig, 2024 | |
| Coot | Emsley et al, 2010 | |
| Phenix | Liebschner et al, 2019 | |
| **Other** | | |
| BacTiter-Glo™ Microbial Cell Viability Assay | Promega GmbH | G8231 |
| Tecan Infinite M200 Pro | Tecan | |
| Octet R2 | Sartorius | |
| Octet NTA Biosensors | Sartorius | 18-5102 |
| ChemiDoc MP Universal Hood III | BioRad | |
| UltiMate 3000 nano-LC system | Thermo Fisher Scientific | |
| Q Exactive HF-X mass spectrometer | Thermo Fisher Scientific | |

## Cloning

The complete list of DNA constructs and primers used in this study is provided in Appendix Table S2.

The pBAD-eGFP (here referred to as pBAD-6xHis-TEV-eGFP) vector was obtained from Michael Davidson (Addgene plasmid #54762). The pBAD-6xHis-TEV and pBAD-6xHis-TEV-eGFP-peptide were obtained by site-directed mutagenesis of pBAD-eGFP (Klimecka et al, 2021). The constructs encoding Myc-degrader fusion peptides were obtained by multistep site-directed mutagenesis by substitution of 6xHis-TEV sequence of pBAD-6xHis-TEV-eGFP with Myc tag and then introducing the degrader-encoding sequence in place of eGFP. The plasmids were amplified by PCR with primers introducing the mutations. The purified products were subjected to phosphorylation and ligation, and the obtained constructs were transformed into *E. coli* Top10 for selection on Ampicillin.

The preparation of constructs for protease purification (pET28a-6xHis-SUMO-ClpP, pET28a-6xHis-SUMO-SspB, pBAD-6xHis-SUMO-ClpX) was described before (Klimecka et al, 2021).

The construct for GroEL purification was obtained by SLIC cloning of GroEL gene amplified from *E. coli* BW25113 DNA into pET28a-6xHis-SUMO vector followed by transformation to *E. coli* Top 10 for selection on Kanamycin. The obtained pET28a-6xHis-SUMO-GroEL was then subjected to 6xHis-SUMO deletion by site-directed mutagenesis as described above to obtain the untagged pET28a-GroEL construct and transformed into *E. coli* Top10 for selection on Kanamycin.

The constructs for peptide purification were obtained by annealing peptide-encoding DNA oligomers, digestion with BamHI and XhoI restriction enzymes, and ligation with digested and purified pET28a-6xHis-SUMO vector followed by transformation to *E. coli* Top10 and selection on Kanamycin.

The construct for 6xHis-ZBD purification was obtained by SLIC cloning of the 6xHis-SUMO-ClpX(1-52) fragment amplified from the pET28a-6xHis-SUMO-ClpX vector (Klimecka et al, 2021) template into pET28a, followed by removal of SUMO by site-directed mutagenesis. The plasmid was amplified by PCR with primers introducing SUMO deletion. The purified PCR product was subjected to phosphorylation and ligation, and the obtained pET28a-6xHis-ZBD construct was transformed into *E. coli* Top10 for selection on Kanamycin.

The construct for purification of His-tagged anti-GFP-nanobody (cAbGFP) was obtained by restriction digestion of a commercially synthesised plasmid (Thermo Fisher Scientific) with a codon-optimised sequence of cAbGFP flanked with restriction sites for XhoI and HindIII enzymes. The insert was subsequently purified and ligated into pBAD-6xHis-TEV plasmid, followed by transformation to *E. coli* Top10 for selection on Ampicillin.

The list of the plasmids with respective oligonucleotides used for cloning is provided in Appendix Table S3. The list of *E. coli* strains used in the study is provided in "Reagents and Tools Table".

## Protein purification

ClpP, ClpX, SspB and eGFP-ssrA were purified as described before (Klimecka et al, 2021).

Overexpression of untagged GroEL was carried out in *E. coli* C43 DE3, overexpression of ZBD domain and SUMO-tagged peptides was carried out in *E. coli* BL21 DE3, and overexpression of cAbGFP was carried out in *E. coli* J42W07. Protein expression from pET28a-based vectors was induced at $OD_{600}$ ~0.6–0.7 with 1 mM IPTG, and protein expression of cAbGFP was induced by the

addition of 0.01% L-arabinose. After 3 h at 30 °C (GroEL and cAbGFP) or 16 h at 18 °C (peptides and ZBD) cultures were centrifuged (5180 × g, 30 min, 4 °C) and bacterial pellets were collected and frozen at −20 °C. Bacterial pellet containing untagged GroEL was lysed in the buffer containing 20 mM Tris pH 7.5, 10 mM NaCl, 10 mM MgCl₂, 1 mM EDTA, 1 mM DTT with the addition of lysozyme (100 mg) and DNase (20 U). After 30 min at 4 °C, the lysate was sonicated (15 min, 45 s on/15 off, 40% amplitude at 4 °C) and centrifuged (48,380 × g, 30 min, 4 °C). Next, the lysate was cleared by filtration (0.8 μm) and applied on the HiTrapQ, 5 ml column (Cytiva). After washing, the column was eluted with a lysis buffer with 1 M NaCl. Fractions containing GroEL identified by SDS-PAGE were combined and then concentrated in Amicon Ultra-15, 30K MWCO (Merck Millipore) (5000 × g at 4 °C). Then, a sample with the addition of 5 mM ATP was applied on a Superdex200 10/300 column (Cytiva) using 50 mM Tris pH 7.5, 150 mM NaCl, 1 mM EDTA, 1 mM DTT, 20% EtOH. Quality of eluted GroEL was assessed on SDS-PAGE, and proper fractions were combined and concentrated in Vivaspin 6, 30K MWCO (Sartorius) (8000 × g at 4 °C) with a buffer change to 50 mM Tris pH 7.5, 150 mM NaCl, 1 mM DTT, 2.5 mM ATP. GroEL concentration was measured at 280 nm, and it was stored at −20 °C.

Bacterial pellets after peptides overexpression were lysed in the buffer containing 25 mM Tris pH 7.5, 300 mM NaCl, 20 mM imidazole, 1 mM DTT with the addition of lysozyme (100 mg) and DNase (20 U). After 30 min at 4 °C, the lysates were sonicated (15 min, 45 s on/15 s off, 40% amplitude at 4 °C) and centrifuged (48,380 × g, 30 min, 4 °C). Next, the lysates were cleared by filtration (0.8 μm) and applied on the on HisTrap, 5 ml column (Cytiva). After washing, the column was eluted with a lysis buffer containing 500 mM imidazole. Presence of peptides was confirmed by SDS-PAGE and proper fractions were combined and dialysed to buffer containing 25 mM HEPES pH 8.0, 300 mM NaCl, 1 mM DTT. After 2 h at 4 °C with one change of buffer peptides were concentrated in Vivaspin 500, 10K MWCO (Sartorius) (10,000 × g at 4 °C), and their concentration was measured at 280 nm. Finally, peptides were flash frozen in liquid nitrogen and stored at −80 °C.

Bacterial pellets with expressed ZBD domain were lysed in the buffer containing 50 mM Tris pH 8.0, 300 mM NaCl, 20 mM imidazole with the addition of lysozyme (100 mg) and DNase (20 U). After 20 min of incubation at 4 °C, the lysates were homogenised (26 psi, 3 rounds at 4 °C) and centrifuged (48,380 × g, 30 min, 4 °C). Next, the lysates were cleared by filtration (0.8 μm) and applied to the HisTrap, 5 ml column (Cytiva). After washing, the column was eluted with a lysis buffer containing 500 mM imidazole. Presence of ZBD domain was confirmed by SDS-PAGE and proper fractions were combined and concentrated in Amicon™ Ultra-15, MWCO 3 kDa, 5000 × g, 4 °C and applied on Superdex200, 10/300 column (Cytiva) using buffer containing 25 mM HEPES pH 8.0, 200 mM KCl, 5 mM MgCl₂, 10% glycerol. Quality of eluted ZBD domain was assessed on SDS-PAGE, and proper fractions were combined, ZBD concentration was measured at 280 nm, and it was flash frozen in liquid nitrogen and stored at −80 °C.

cAbGFP bacterial pellet was lysed in the buffer containing 50 mM Tris 8.0, 150 mM NaCl, 30 mM imidazole, 1 mM DTT, 1 mM PMSF with the addition of lysozyme (100 mg) and DNase (20 U). After 30 min of incubation at 4 °C, the lysates were sonicated (15 min, 45 s on/15 off, 40% amplitude at 4 °C) and centrifuged (48,380 × g, 30 min, 4 °C). Next, the lysate was cleared by filtration (0.8 μm) and applied to the HisTrap, 5 ml column (Cytiva). After washing, the column was eluted with a lysis buffer with 300 mM imidazole. After SDS-PAGE analysis, fractions containing cAbGFP were combined and concentrated in Amicon™ Ultra-6, MWCO 10K, 4000 × g, 4 °C and applied on Superdex75 10/300 column (Cytiva) using buffer containing 20 mM HEPES pH 8.0, 150 mM NaCl, 1 mM DTT. Quality of eluted cAbGFP was assessed on SDS-PAGE, and proper fractions were combined, concentration was measured at 280 nm, and protein was flash frozen in liquid nitrogen and stored at −80 °C.

## eGFP degradation in bacteria

The degradation of eGFP-peptide constructs in bacteria was performed as described before (Klimecka et al, 2021) with minor modifications. Overnight cultures of E. coli BW25113 or mutant strains (Baba et al, 2006) carrying plasmids: pBAD-6xHis-TEV, pBAD-6xHis-TEV-eGFP, pBAD-6xHis-TEV-eGFP-ssrA, pBAD-6xHis-TEV-eGFP-AANDENY, pBAD-6xHis-TEV-eGFP-2xAAN-DENY, pBAD-6xHis-TEV-eGFP-XB, and pBAD-6xHis-TEV-eGFP-sXB were grown in LB medium supplemented with Ampicillin (100 μg/ml) at 37 °C with shaking. The cultures were then diluted 1:100 in 5 ml of fresh LB with Ampicillin (100 μg/ml) and grown to mid-exponential phase (OD₆₀₀ 0.5–0.6). The expression of plasmids was induced by the addition of L-arabinose to a final concentration of 0.005%, and the bacteria were further cultured overnight at 18 °C. The cultures were then diluted 1:20 in a 96-well microplate in 200 μl of M9 medium supplemented with Ampicillin (100 μg/ml), Spectinomycin (100 μg/ml) and 0.2% glucose (to arrest the expression of protein constructs). The degradation was then monitored by measuring OD₆₀₀ and fluorescence (excitation at λ = 489 nm and emission at λ = 520 nm). The measurements were taken every 15 min for 6 h at 30 °C with shaking in between measurements in Tecan Infinite M200 Pro plate reader. The fluorescence was normalised to the optical density, and the background of bacteria expressing the control pBAD-6xHis-TEV vector was subtracted. The values at 0 time point were set as 100%.

The bortezomib inhibition assay was performed likewise, except that after the overnight induction the bacterial cultures were diluted in the antibiotics-supplemented M9 medium with the addition of 1% DMSO or Bortezomib in DMSO in concentrations of 10 μM, 50 μM, 100 μM, 250 μM or 500 μM.

## Serial dilution plate test

Overnight cultures of E. coli BW25113 strains transformed with pBAD plasmids encoding degraders and control peptides were grown in LB medium with Ampicillin (100 μg/ml). The cultures' OD₆₀₀ was measured and normalised to 1.0. The cultures were then serially diluted in sterile PBS in final concentrations from 10⁻¹ to 10⁻⁶. The 2 μl drops of the dilutions were put on LB agar with Ampicillin and with or without 0.02% L-arabinose. The plates were then incubated in appropriate temperatures overnight (for temperatures 37 °C and above) or up to 36 h (for lower temperatures).

## Bacterial growth measurement

Overnight cultures of *E. coli* BW25113 or mutant strains transformed with pBAD plasmids encoding degraders and control peptides were grown in LB medium with Ampicillin (100 µg/ml) and in case of mutant strains also Kanamycin (30 µg/ml). The cultures were then diluted 1:50,000 in a 96-well microplate in 200 µl of LB supplemented with appropriate antibiotics and with or without the addition of 0.02% L-arabinose. The cell growth was then monitored by measuring $OD_{600}$. The measurements were taken every 30 min for 16 h at either 30 °C or 42 °C with shaking in between measurements in the Tecan Infinite M200 Pro plate reader. The dose-response experiments were performed likewise, except the fivefold dilutions of arabinose were used, ranging from 0.1% to $6.4e^{-6}$ %.

## Measuring bacterial viability with BacTiter-Glo

Overnight cultures of *E. coli* BW25113 strains transformed with pBAD plasmids encoding degraders and control peptides were grown in LB medium with Ampicillin (100 µg/ml). The cultures were then diluted 1:100 in 10 ml of fresh LB with Ampicillin and grown at 37 °C to $OD_{600}$ 0.1–0.2. The cultures were then induced with 0.02% L-arabinose and grown at 30 or 42 °C with shaking. The $OD_{600}$ and bacterial viability measurements were taken at 0, 2, 4 and 6 h post-induction. The bacterial viability was measured using luminescence-based BacTiter-Glo Microbial Viability Assay (Promega) according to the manufacturer's protocol. The luminescence measurements were taken in a Tecan Infinite M200 Pro plate reader.

## Western blot

Overnight cultures of *E. coli* BW25113 strains transformed with pBAD plasmids encoding degraders and control peptides were grown in LB medium with Ampicillin (100 µg/ml). The cultures were then diluted 1:100 in 10 ml of fresh LB with Ampicillin and grown in 37 °C to $OD_{600}$ 0.1–0.2. The cultures were then induced with 0.02% L-arabinose, and the temperature was shifted to 42 °C. The samples corresponding to $OD_{600} = 0.2$ were collected at 0, 1, 2 and 6 h after induction. The culture aliquots were centrifuged for 5 min at $2000 \times g$ at 4 °C, washed once in cold PBS, and the pellets were stored at -80 °C until further processing. The samples were suspended in a Laemmli sample buffer, and samples corresponding to $OD_{600} = 0.05$ were run on 10% SDS-PAGE gels with 0.5% 2,2,2-trichloroethanol (Ladner et al, 2004). The gels were then activated for 5 min by 300 nm irradiation on ChemiDoc Imaging System (Bio-Rad), and then the proteins were transferred on 0.4 µm PVDF membrane in Tris-Glycine buffer with 20% methanol for 90 min at 350 mA with cooling. The membranes were then blocked in 3% BSA in TBST buffer for 1 h in room temperature with agitation followed by overnight incubation with 1:2500 mouse anti-GroEL antibody (Invitrogen MA3-022) and 1:10,000 rabbit anti-enolase (a gift from prof. Ben Luisi) in 3% BSA in TBST overnight at 4 °C. The following day the membranes were washed three times in TBST and incubated with 1:10,000 anti-mouse Alexa Fluor 647 and 1:10,000 anti-rabbit Alexa Fluor 488 antibodies (Thermo Fisher Scientific) in 3% BSA in TBST for 1 h at room temperature. The blots were then scanned using the ChemiDoc Imaging System

(Bio-Rad) and band intensities were quantified in Image Lab Software (Bio-Rad).

## Mass spectrometry proteomic measurement with isobaric labelling (TMT-MS)

Overnight cultures of *E. coli* BW25113 strain transformed with pBAD plasmids encoding degraders and control peptides were grown in LB medium with Ampicillin (100 µg/ml). The cultures were then diluted 1:100 in 10 ml of fresh LB with Ampicillin in triplicates and grown in 37 °C to $OD_{600}$ 0.1–0.2. The cultures were then induced with 0.02% L-arabinose and grown for 1, 2 or 6 h at 42 °C with shaking. The cultures were then centrifuged for 10 min at $1000 \times g$ at 4 °C, washed once in cold PBS, and the pellets were stored at −80 °C until further processing.

Proteins from bacterial pellets were extracted using the Sample Preparation by Easy Extraction and Digestion (SPEED) protocol (Doellinger et al, 2020). In brief, bacterial pellets were solubilised in concentrated TFA (cell pellet/TFA 1:2-1:4 (v/v)) and incubated for 2–10 min at RT. Samples were neutralised by adding 2 M Tris-Base buffer using 10× volume of TFA and further incubated at 95 °C for 5 min after adding Tris(2-carboxyethyl)phosphine (TCEP) to a final concentration of 10 mM and 2-Chloroacetamide (CAA) to a final concentration of 40 mM. Protein concentrations were determined by turbidity measurements at 360 nm, adjusted to the same concentration using a sample dilution buffer (2 M Tris Base/TFA 10:1 (v/v)) and then diluted 1:4–1:5 with water. Digestion was carried out overnight at 37 °C using trypsin (sequencing grade, Promega) at a protein/enzyme ratio of 100:1. TFA was added to a final concentration of 2% to stop digestion. The resulting peptides were labelled using an on-column TMT labelling protocol (Myers et al, 2019). TMT-labelled samples were compiled into a single TMT sample and concentrated using a SpeedVac concentrator. Peptides in the compiled sample were fractionated (6 or 8 fractions) using the bRP fractionation. Prior to LC-MS measurement, the peptide fractions were resuspended in 0.1% TFA, 2% acetonitrile in water.

Chromatographic separation was performed on an Easy-Spray Acclaim PepMap column 50 cm long × 75 µm inner diameter (Thermo Fisher Scientific) at 55 °C by applying 90–120 min acetonitrile gradients in 0.1% aqueous formic acid at a flow rate of 300 nl/min. An UltiMate 3000 nano-LC system was coupled to a Q Exactive HF-X mass spectrometer via an easy-spray source (all Thermo Fisher Scientific). The Q Exactive HF-X was operated in TMT mode with survey scans acquired at a resolution of 60,000 at $m/z$ 200. Up to 15 of the most abundant isotope patterns with charges 2–5 from the survey scan were selected with an isolation window of 0.7 m/z and fragmented by higher-energy collision dissociation (HCD) with normalised collision energies of 32, while the dynamic exclusion was set to 30 or 45 s. The maximum ion injection times for the survey scan and the MS/MS scans (acquired with a resolution of 45,000 at $m/z$ 200) were 50 and 96 or 150 ms, respectively. The ion target value for MS was set to 3e6 and for MS/MS to 1e5, and the minimum AGC target was set to 1 or 2e3.

The data were processed with MaxQuant v. 1.6.17.0 (Tyanova et al, 2016a), and the peptides were identified from the MS/MS spectra searched against Uniprot *E. coli* Reference Proteome (UP000000625) supplemented with sequences of GroTAC and control peptides using the built-in Andromeda search engine.

Reporter ion MS2-based quantification was applied with reporter mass tolerance = 0.003 Da and min. reporter PIF = 0.75. Cysteine carbamidomethylation was set as a fixed modification and methionine oxidation as well as glutamine/asparagine deamination were set as variable modifications. For in silico digests of the reference proteome, cleavages of arginine or lysine followed by any amino acid were allowed (trypsin/P), and up to two missed cleavages were allowed. The FDR was set to 0.01 for peptides, proteins and sites. Match between runs was enabled. Other parameters were used as pre-set in the software.

Protein Groups table was loaded into Perseus v.1.6.10.0 (Tyanova et al, 2016b). Standard filtering steps were applied to clean up the dataset: reverse (matched to decoy database), only identified by site, and potential contaminants (from a list of commonly occurring contaminants included in MaxQuant) protein groups were removed. Reporter intensity corrected values were log2 transformed and protein groups with all 15 values in a given sample set were kept. The intensity values were then normalised by median subtraction within TMT channels. Student T-testing (permutation-based FDR = 0.01, S0 = 0.2) was performed on the dataset to return protein groups, which levels were statistically significantly changed between groups of samples. The results are provided in a Dataset EV1.

This dataset has been deposited to the ProteomeXchange Consortium via the PRIDE partner repository with the dataset identifier PXD045730 (Perez-Riverol et al, 2016, 2022; Deutsch et al, 2023).

## ATPase activity assay

The ATPase activity assay was performed as described before (Klimecka et al, 2021) using an NADH-coupled assay (Nørby, 1988). The assay was performed in a 96-well plate and carried out in a buffer containing 50 mM Tris (pH 7.5), 200 mM NaCl and 5 mM $MgCl_2$ in the presence of 1–2 mM NADH, 2.5 mM ATP, 2.5 mM phosphoenolpyruvate, 50 μg/ml pyruvate kinase, and 50 μg/ml lactate dehydrogenase. The ClpX interactors (SspB or researched peptides) were added in the appropriate concentrations, and the reaction was started by adding 0.08 μM of $ClpX_6$ to the final volume of 100 μl. Changes in NADH concentration were measured via changes in the absorbance at 340 nm at 30 °C. The absorbance was monitored for 1.5–2 h at 1 min intervals in a Tecan Infinite 200 Pro plate reader (Tecan Group Ltd.). The rate of ATP hydrolysis was calculated assuming a 1:1 ratio between ATP regeneration and NADH oxidation and a $\Delta\varepsilon340$ of 6.23 $\mu M^{-1} cm^{-1}$. Within each run, the values corresponding to the same interactor concentration were averaged and treated as a technical replicate. Curve fitting and statistical analysis were performed via GraphPad Prism 9 software (GraphPad Software, LLC). The error bars representing SEM are plotted for the points with more than one technical replicate.

## SDS-PAGE analysis of in vitro degradation

SDS-PAGE analysis of in vitro GroEL degradation by ClpXP was performed as described previously (Klimecka et al, 2021). Reactions were performed in a buffer containing 50 mM Tris (pH 7.5), 200 mM NaCl, 10 mM KCl, and 10 mM $MgCl_2$ in the presence of 2.5 mM ATP, 2.5 mM phosphoenolpyruvate, and 50 μg/ml pyruvate kinase. GroEL or control protein was added to the buffer in the

concentration 0.1 μM of $GroEL_{14}$ incubated in RT with or without the presence of 35 μM peptide (XB-GGS, XB-GGS-SBP, or GGS-SBP). After 10 min, 0.8 μM of $ClpX_6$ was added, and the samples were further incubated for 15 min at RT. The reaction was started by adding 2.1 μM of $ClpP_{14}$ to the final volume of 100 μl and performed at 30 °C. In total, 10 μl samples were collected in 2 h intervals up to 10 h. The degradation reaction was stopped by adding the Laemmli loading buffer, followed by 5 min incubation at 95 °C. Prepared samples were applied on 8–10% SDS-PAGE gels, resolved at 200 V and visualised using SafeStain dye. The band intensities were measured in Image Lab Software (Bio-Rad). The intensities of bands corresponding to GroEL were normalised to the intensities of the pyruvate kinase bands. The relative intensity at 0 h was treated as 100%, and the rest of the points were compared to the point at 0 h.

## Characterisation of peptide-protein interactions by biolayer interferometry

The interactions of His-SUMO-tagged peptides (GroTAC1, XB-GGS, GGS-SBP) with ClpX and GroEL were characterised by biolayer interferometry using Octet R2 (Sartorius) and Octet NTA Biosensors. The optimal protein and peptide concentrations were determined separately for each of the proteins by performing ligand scout experiments at fixed protein concentration and different ligand concentrations, including sensors without ligands as binding specificity control. The interactions between ClpX and the His-SUMO-tagged peptides were measured at 30 °C in the buffer containing 25 mM HEPES (pH 8.0), 200 mM KCl, 10 mM $MgCl_2$, 0.05 % Tween-20, and 2.5 mM ATP. The sensors were loaded with 150 nM of His-SUMO-XB-GGS (Anchor-Linker) or 100 nM of His-SUMO-XB-GGS-SBP (GroTAC1), followed by blocking with 86 nM His-SUMO. The binding kinetics was then determined by measuring the interaction with a two-fold dilution series of ClpX in concentrations from 1 μM to 31.25 nM. A control without ClpX was included in the assay. The interaction with GroEL was measured at 30 °C in the buffer containing 25 mM HEPES (pH 8.0), 300 mM KCl, 10 mM $MgCl_2$, 0.05% Tween-20, and 12.5 mM ATP. The sensors were loaded with 200 nM His-SUMO-GGS-SBP (Linker-Bait) or 400 nM His-SUMO-XB-GGS-SBP (GroTAC1) and blocked with 257 nM His-SUMO. The concentration range of GroEL ranged from 2.5 μM to 78.1 nM with one reference sensor without GroEL. The results were analysed with the Octet Analysis Studio software (Sartorius), subtracting the signal from the sensor without the analyte and aligning the initial baseline and dissociation steps. The obtained kinetic parameters were obtained assuming the 1:1 binding ratio.

The interactions of untagged, synthetic peptides (GroTAC1, GroTAC2, GroTAC3, Appendix Table S4) with ZBD were characterised using the Octet NTA Biosensors. The optimal ZBD concentration was determined by performing loading scout experiments. The interaction was measured at 30 °C in the buffer containing 25 mM HEPES (pH 8.0), 200 mM KCl, 10 mM $MgCl_2$, 2.5 mM ATP and 0.05% Tween 20. The sensors were loaded with 1.37 μM His-ZBD followed by blocking with 600 nM cAbGFP (a His-tagged nanobody targeting GFP). The binding kinetics were assessed by measuring the interaction between ZBD and with a twofold dilution series of peptides ranging from 1 μM to 31.25 nM. The interaction of each GroTAC concentration was measured as a

reference on a sensor not loaded with ZBD. As an additional reference, the control without GroTAC was included in the assay. The results were analysed using Octet Analysis Studio software (Sartorius), subtracting both reference signals from the sensors loaded with GroTAC and aligning the initial baseline and dissociation steps. The kinetic parameters were obtained assuming the 1:1 binding ratio.

The ternary complex formation between synthetic peptides (GroTAC1, GroTAC2, Linker-Bait, Appendix Table S4) and GroEL with ZBD protein were measured at 30 °C in the buffer containing 25 mM HEPES (pH 8.0), 200 mM KCl, 10 mM MgCl$_2$, 2.5 mM ATP and 0.05% Tween-20. The sensors were loaded with 1.37 μM His-ZBD followed by blocking with 600 nM cAbGFP. The formation of the ternary complex was demonstrated by comparing the interaction of peptides (GroTAC1, GroTAC2, XB, SBP) at 1 μM with the interaction of the same peptides pre-incubated with GroEL (1 μM) for 20 min at 30 °C. The binding kinetics were assessed by measuring the interaction between ZBD and a mix of a twofold dilution series of peptides (GroTAC1, GroTAC2) ranging from 1 μM to 31.25 nM, with a fixed concentration of GroEL (1 μM). The interaction of each GroTAC concentration without GroEL was measured as a reference on a sensor not loaded with ZBD. As an additional reference, the interaction of GroEL alone (1 μM) with ZBD was assessed. The results were analysed using Octet Analysis Studio software (Sartorius), subtracting both reference signals from the sensors loaded with the GroTAC-GroEL mixture and aligning the initial baseline and dissociation steps. The kinetic parameters were obtained using the responses from the three highest concentrations assuming the 1:1 binding ratio.

## Cryo-EM structure of the GroEL-GroTAC complex

The complex of GroEL with chemically synthesised GroTAC3 (XB-PEG3-SBP) peptide (KareBay Biochem) was formed by mixing 5 μM GroEL$_{14}$ with 140 μM peptide in the Cryo-EM buffer (50 mM HEPES pH 8.0, 200 mM KCl, 5 mM MgCl$_2$) and incubating the mixture in room temperature for 20 min. The ATP was added immediately before freezing the sample for the final concentration of 1 mM. In all, 3 μl of the sample mixture was deposited on the glow discharged (25 mA, 0.38 mbar for 50 s) Quantifoil R 2/2 mesh 200 Cu grid and vitrified in liquid ethane with an FEI Vitrobot Mark IV (Thermo Fisher Scientific) at 4 °C with 100% humidity and 3 s blot time with blotting force 3. The grid was imaged with a Glacios electron microscope (Thermo Fisher Scientific) that operated at 200 kV and was equipped with a Falcon 3EC camera at the Cryomicroscopy and Electron Diffraction Core Facility at the Centre of New Technologies, University of Warsaw. A total of 4672 movies in.mrc format were recorded in a counting mode with a physical pixel size of 0.5878 Å (nominal magnification of ×240,000), 100 μm objective aperture, and nominal defocus range of −2.2 to −0.8 μm (with 0.2-μm steps). The total dose (fractionated into 27 frames) was 40 e/Å$^2$, and the dose rate was 0.76 e/pixel/s.

## Cryo-EM data processing

Cryo-EM images were processed with cryoSPARC 4.2. First, raw movies were subjected to the cryoSPARC in-built patch motion correction followed by the patch CTF estimation algorithms. A total of 350,383 particles were picked from the whole dataset using a template picker generated by a manual picker. Particles were subjected to four rounds of 2D classification with 250 Å circular mask diameter. A total of 203,031 particles were extracted with box size of 560 px and used to generate de novo a 3D initial model with maximum resolution set to 12 Å. The generated model was aligned to D7 symmetry axes and 3D refined with imposed D7 symmetry using the homogenous refinement algorithm, which resulted in 2.91 Å reconstruction. The model was then used to run a Reference-Based Motion Correction in cryoSPARC 4.4. The resulting particles were used for a new Ab Initio model followed by homogenous refinement with imposed D7 symmetry, as well as local and global CTF refinements and the Ewald sphere correction with negative curvature sign, which produced a map with a resolution of 2.45 Å. The obtained map was deposited in EMDB (ID EMD-19687). (An asymmetric reconstruction was also tried, but it provided poorer map density for the GroTAC ligand and worse modelling results, so that it was not used for preparation of an alternative final model). In order to visualise relative local resolution of the cryo-EM map, we used OccuPy with default parameters (Forsberg et al, 2023).

## Model building in cryo-EM maps

The obtained Cryo-EM maps were processed using a few standard modelling protocols for building models from Cryo-EM data. A portion of GroEL was usually reconstructed in agreement with available structures in the PDB database; however, due to the lower resolution of the peptide fragments, the interacting peptides were not reconstructed. Therefore, we employed the newly published ModelAngelo method, which automates atomic modelling in Cryo-EM maps and constructs protein models of comparable quality to those built by human experts (Jamali et al, 2024). ModelAngelo utilises machine learning to model building in three steps: (1) prediction of residues (Cα atoms positions) based on cryo-EM map, (2) optimisation of residue positions and orientations, creating a graph of connections, (3) generating of a full-atom model.

In many GroEL regions, especially near GroEL-GroTAC interface and in the flexible loops on the outside of the barrel, the map had much lower resolution. Consequently, the neural network predicted different residues than the input sequence of GroEL, which were then removed from the final structure in the 3rd stage of ModelAngelo refinement. Therefore, we decided to use the model from the previous step (in Cα representation) for further processing. Gaps (errors) in the identification of GroEL residues were manually corrected. Based on available PDB structures (PDB ID 1MNF), we changed the identity of misidentified residues. Prediction of peptide fragments was not straightforward due to the low resolution of the cryo-EM map in this region. In the expected peptide region, ModelAngelo identified usually from 7 to 10 Cα positions (which is much fewer than the 13 residues from the XB peptide, the (PEG)$_3$ linker, and the 12 residues of SBP). In most cases, predicted residues differed significantly from the peptide sequence. One of the only consistently recognised residues was proline at one of the Cα positions. When we superimposed the ModelAngelo structure on the PDB structure (PDB ID 1MNF), this Cα position was near the P6 Cα position from SBP. Therefore, without any other leads, we assigned identities to the Cα positions

in ModelAngelo structure, that were close in the superimposed PDB structure. The comparison between our final model and PDB model can be seen in Appendix Fig. S3. Finally, the obtained Cα resolution model was reconstructed into a full-atom representation using cg2all method guided by cryo-EM data. Cg2all has been demonstrated to be a useful tool in the accurate refinement of all-atom coordinates against intermediate- and low-resolution cryo-EM densities (Heo and Feig, 2024). This model was then used for refinement against the map. The first stage of refinement was done in Coot (Emsley et al, 2010). The final refinement stage involved two rounds of Real Space Refine in Phenix (Liebschner et al, 2019). After final corrections, the obtained model was validated and deposited to the PDB database (PDB ID 8S32).

## Graphics

Figures 1, 5 and the synopsis image graphics were created with BioRender.com (Górna M, 2025) https://BioRender.com/rjrsa12, https://BioRender.com/em7jcym, https://BioRender.com/xh055s6).

# Data availability

The datasets produced in this study are available in the following databases: The proteomics dataset has been deposited to the ProteomeXchange Consortium via the PRIDE partner repository (PXD045730). The cryo-EM map of GroEL-GroTAC3 complex is available in EMDB (ID EMD-19687). The refined model of GroEL-GroTAC3 complex is available in PDB (PDB ID 8S32).

The source data of this paper are collected in the following database record: biostudies:S-SCDT-10_1038-S44319-025-00510-9.

# Peer review information

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

## Acknowledgements

The authors thank Agata Krawczyk-Balska, Ben Luisi and Paulina Dominiak for helpful discussions. This work was funded by grant number POIR.04.04.00-00-5EC1/18-00 to MG for the project "Proteolysis-targeting strategies in bacterial systems for functional studies of proteins and improvement of antibiotics" carried out within the FIRST TEAM programme of the Foundation for Polish Science co-financed by the European Union under the European Regional Development Fund. KW and SK acknowledge funding from the National Science Centre, Poland [2020/39/B/NZ2/01301]. Cryo-EM data collection was enabled by the Ministry of Science and Higher Education programme "Excellence Initiative—Research University (2020-2026) (IDUB)" awarded to the University of Warsaw, in particular, the 1.4.1. Activity "Strengthening the core-facility potential on the Ochota Campus" and grant BOB-IDUB-622-20/2021 entitled "Infrastructure for Cryomicroscopy and Electron Diffraction Core Facility" & I.4.2. Fund for the Renovation and Development of Research Infrastructure—development and maintenance of infrastructure.

## Author contributions

**Matylda Anna Izert-Nowakowska**: Conceptualisation; Formal analysis; Supervision; Validation; Investigation; Visualisation; Methodology; Writing—original draft; Writing—review and editing. **Maria Magdalena Klimecka**: Conceptualisation; Formal analysis; Supervision; Validation; Investigation; Methodology; Writing—original draft; Writing—review and editing. **Anna Antosiewicz**: Formal analysis; Investigation; Methodology. **Karol Wróblewski**: Formal analysis; Investigation; Methodology; Writing—original draft; Writing—review and editing. **Jakub Józef Kowalski**: Formal analysis; Validation; Investigation; Methodology; Writing—original draft; Writing—review and editing. **Katarzyna Justyna Bandyra**: Investigation; Methodology; Writing—review and editing. **Tomasz Góral**: Formal analysis; Investigation; Writing—review and editing. **Sebastian Kmiecik**: Supervision; Funding acquisition. **Remigiusz Adam Serwa**: Formal analysis; Investigation; Writing—review and editing. **Maria Wiktoria Górna**: Conceptualisation; Formal analysis; Supervision; Funding acquisition; Investigation; Visualisation; Methodology; Writing—original draft; Project administration; Writing—review and editing.

Source data underlying figure panels in this paper may have individual authorship assigned. Where available, figure panel/source data authorship is listed in the following database record: biostudies:S-SCDT-10_1038-S44319-025-00510-9.

## Disclosure and competing interests statement

MWG, MAIN, MMK and AA hold a patent application related to this work (P.439032, PCT/IB2022/059128).

# Expanded View Figures

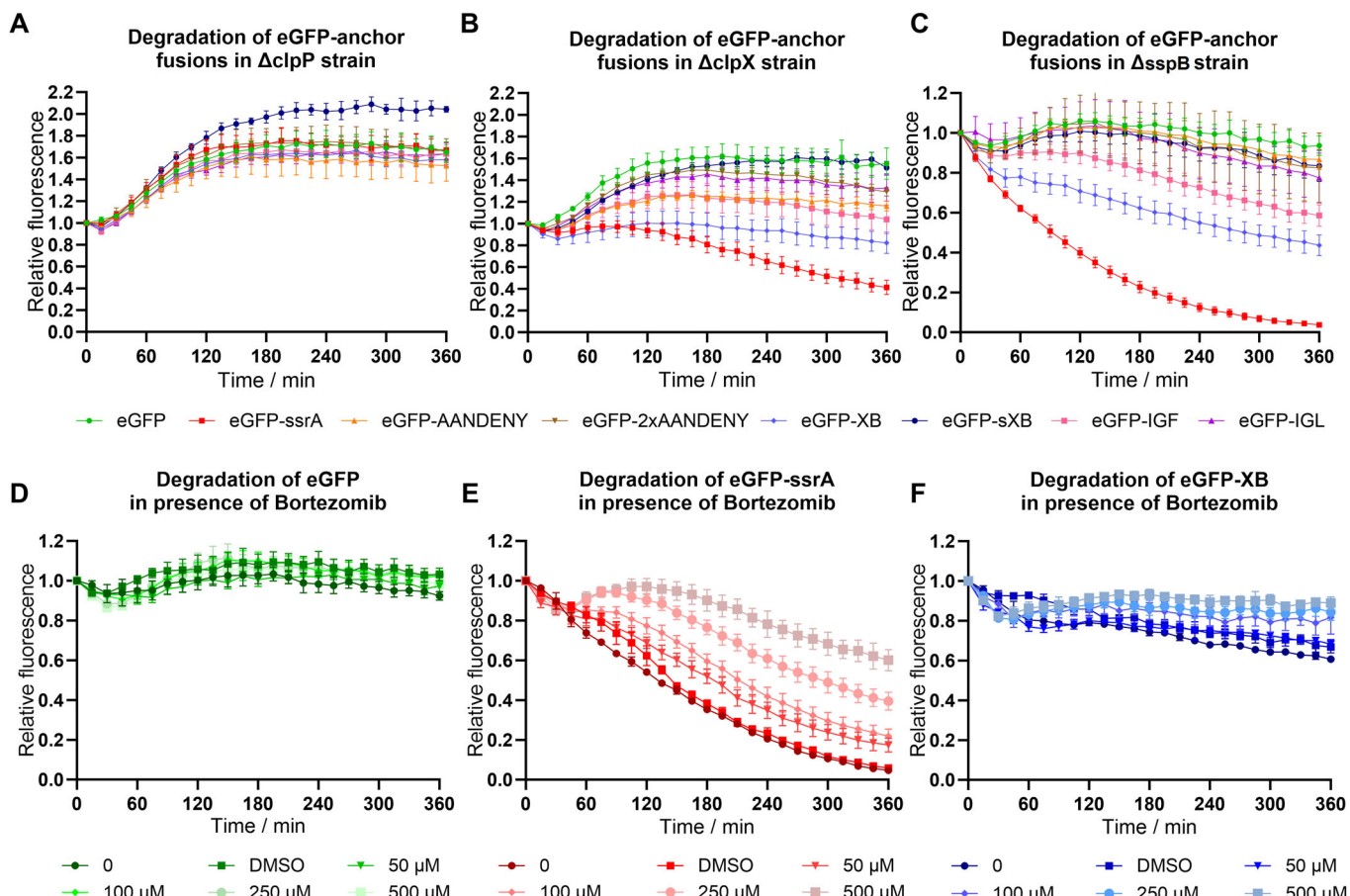

**Figure EV1.  Degradation of the eGFP-anchor fusion proteins in bacteria.**

Degradation was performed in *E. coli* deletion mutant strains (**A**) ΔclpP, (**B**) ΔclpX, and (**C**) ΔsspB. Degradation was performed in *E. coli* BW25113 strain for (**D**) untagged eGFP (negative control), (**E**) eGFP-ssrA (positive control), and (**F**) eGFP-XB. The curves represent mean values from 3 biological repeats (averaged for clarity) with error bars representing SEM. Source data are available online for this figure.

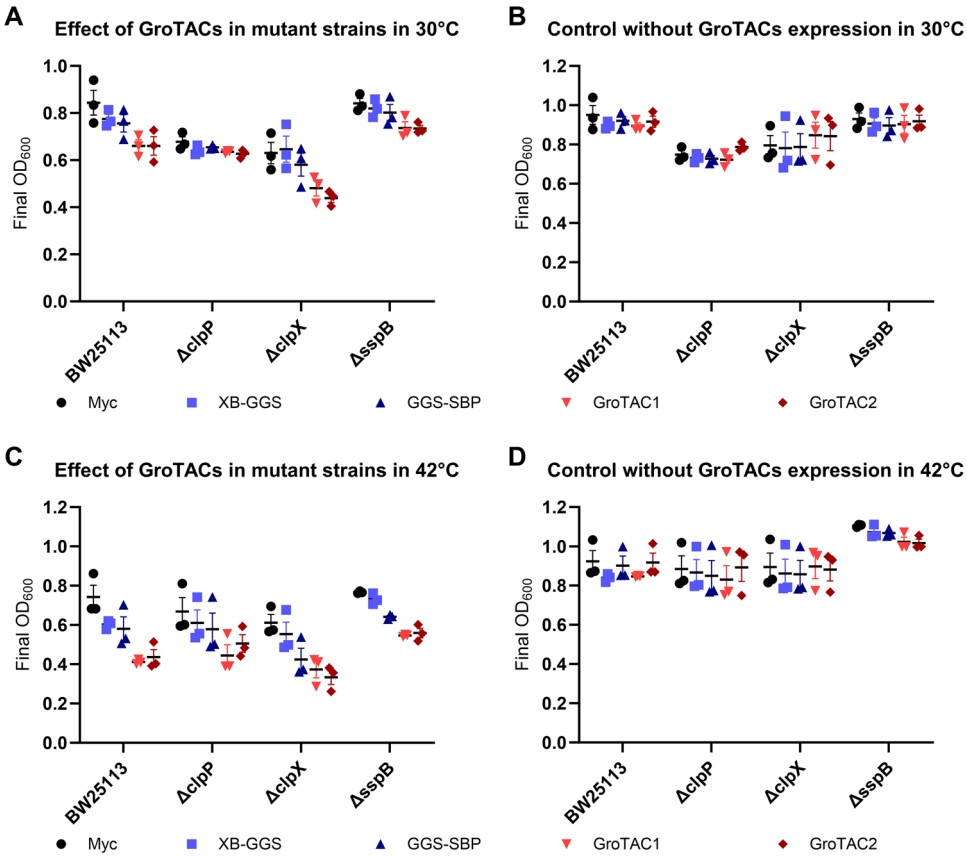

**Figure EV2. The effect of protease component mutations on the GroTACs-mediated growth inhibition.**

The expression of GroTACs was tested in *E. coli* deletion mutant strains for their effect on final culture $OD_{600}$ after 16 h of culturing (**A, C**) in presence, or (**B, D**) in absence of expression-inducing arabinose at 30 °C (**A, B**) or 42 °C (**C, D**). The horizontal lines represent the mean from three biological replicates and the error bars represent SEM. Source data are available online for this figure.

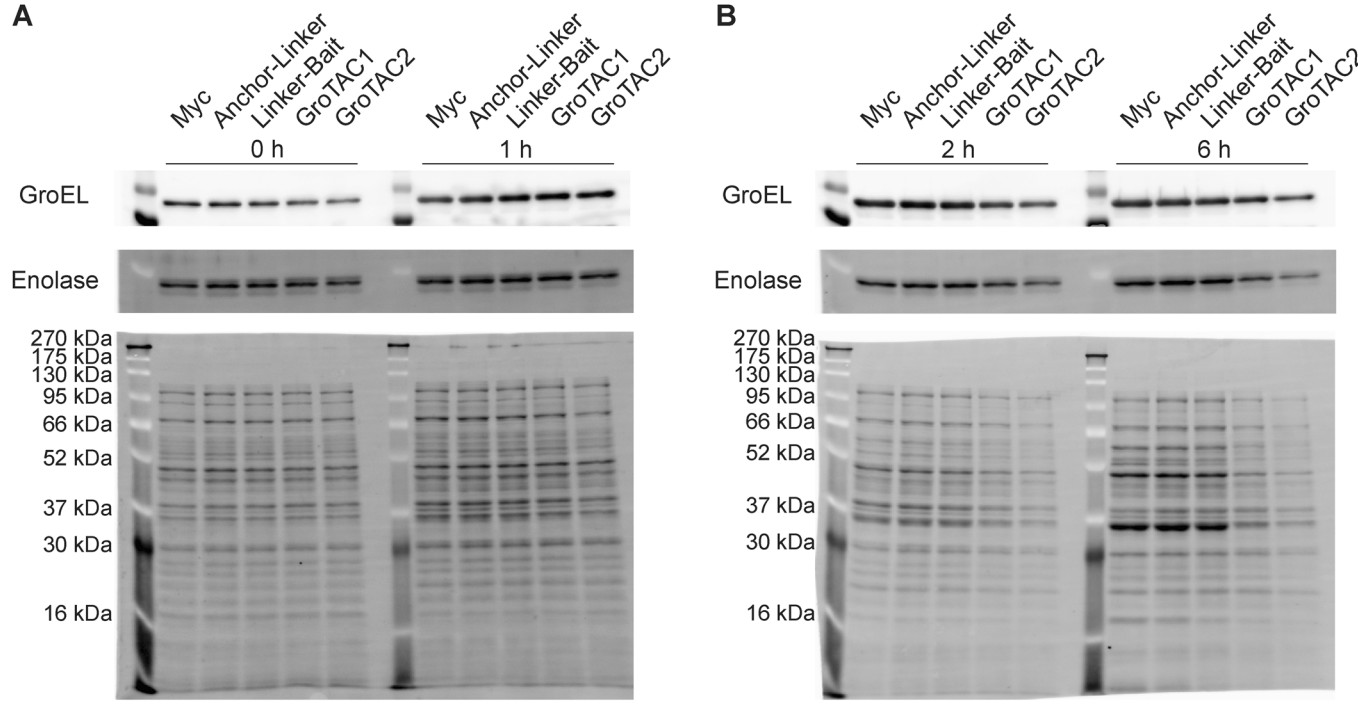

**Figure EV3.  The effect or GroTACs expression on proteins in *E. coli*.**

The figure presents representative western blots of GroEL and enolase levels and total protein on PVDF membranes visualised by the stain-free method (Ladner et al, 2004). Protein levels were measured (**A**) 0 h and 1 h after induction and (**B**) 2 h and 6 h after induction. Source data are available online for this figure.

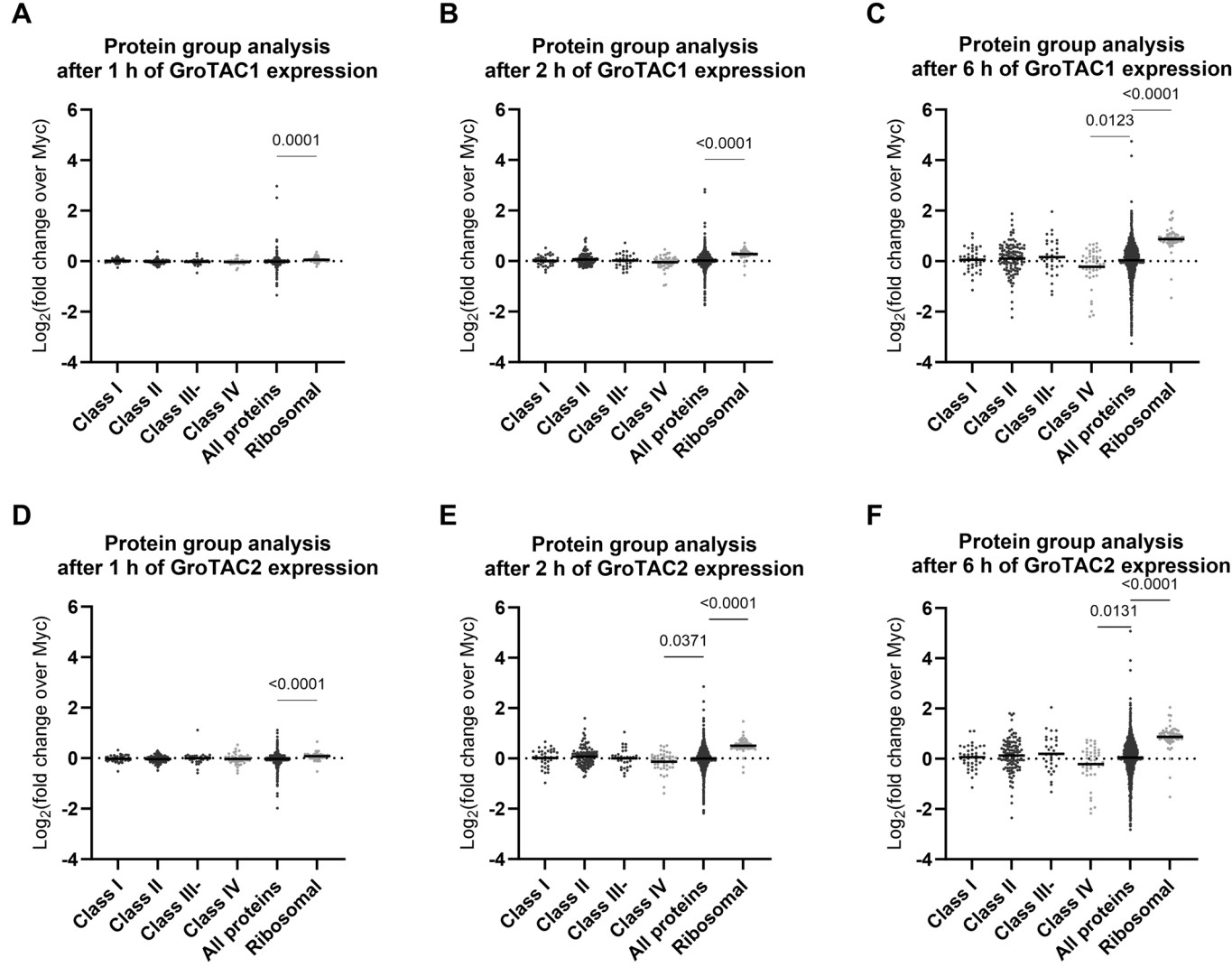

**Figure EV4. GroTAC-induced changes in specific protein groups.**

(Group analysis of relative changes in levels of GroEL substrates and ribosomal proteins induced by GroTAC1 (**A–C**) and GroTAC2 (**D–F**) after 1 h (**A, D**) 2 h (**B, E**) and 6 h (**C, F**) of peptide expression. Statistically significant changes (two-sided, unpaired Student's *t* test) are denoted by *P* values, and the mean value in each group is indicated by a horizontal line. Data for TMT-MS were collected for three biological replicates. Source data are available online for this figure.

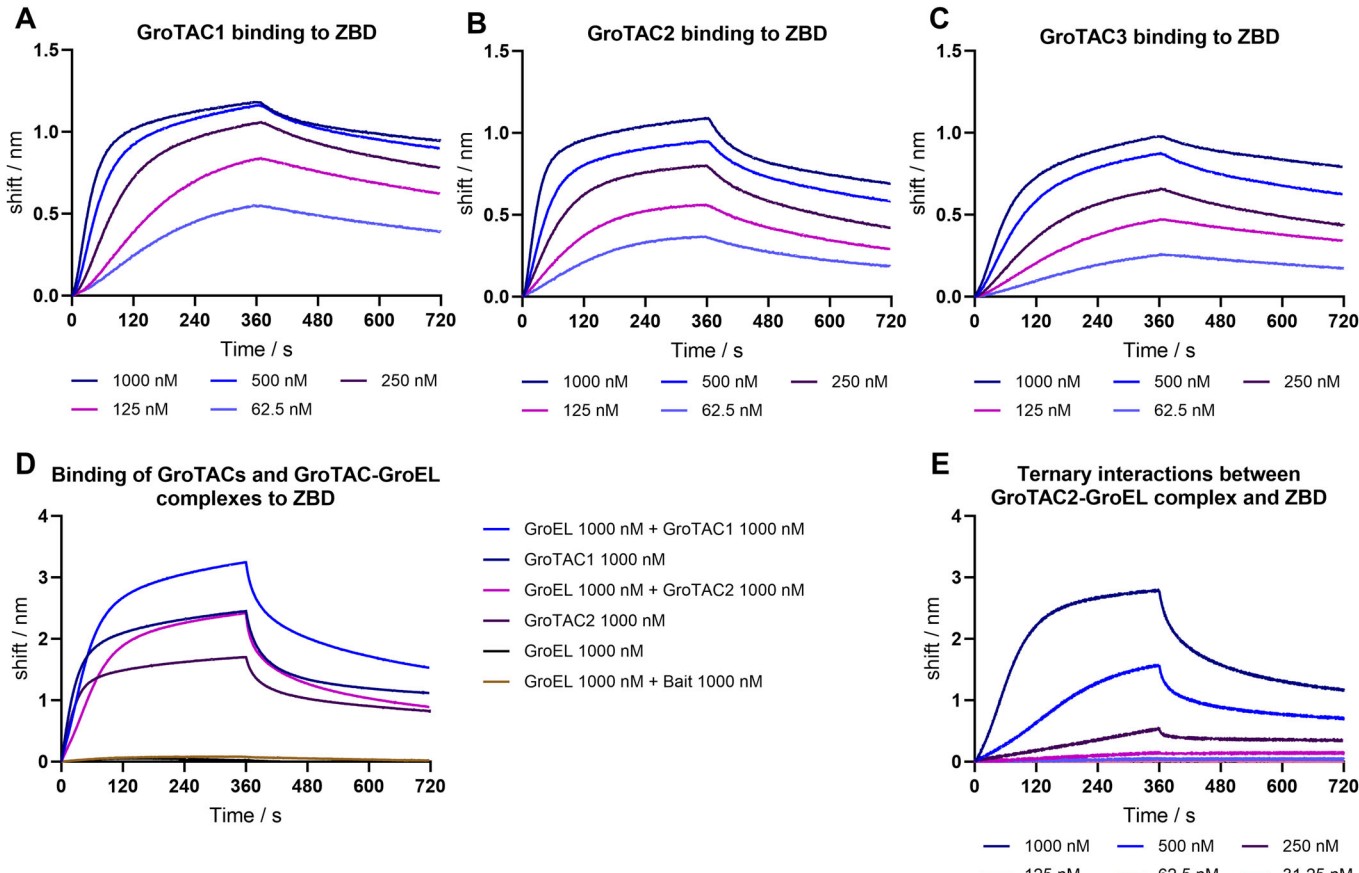

**Figure EV5.  Binding of untagged GroTACs to ZBD.**

Sensograms representing BLI measurements. (A) Binding of the synthetic GroTAC1 peptide to immobilised His-ZBD. (B) Binding of the synthetic GroTAC2 peptide to immobilised His-ZBD. (C) Binding of GroTAC3 to immobilised His-ZBD. (D) Comparison of binding of GroEL, synthetic GroTAC peptides or their complexes to immobilised His-ZBD. "Bait" is a synthetic control peptide consisting of a "Linker-Bait" fusion. (E) Binding of GroEL-GroTAC2 complex to immobilised His-ZBD using a fixed GroEL concentration (1000 nM) and increasing concentrations of the synthetic GroTAC2 peptide in the preincubated GroTAC2-GroEL mixture. Source data are available online for this figure.

