## [Peer Review File · EMBO Reports]

Targeted Protein Degradation in Escherichia coli using CLIPPERS

Matylda Izert-Nowakowska, Maria Klimecka, Anna Antosiewicz, Karol Wróblewski, Jakub Kowalski, Katarzyna Bandyra, Tomasz Góral, Sebastian Kmiecik, Remigiusz Serwa, and Maria Gorna

Corresponding author(s): Maria Gorna (mw.gorna@uw.edu.pl)

Review Timeline:

Submission Date:	11th Apr 24
Editorial Decision:	10th Jun 24
Appeal Received:	19th Sep 24
Editorial Decision:	2nd Oct 24
Appeal Received:	1st Apr 25
Editorial Decision:	30th Apr 25
Revision Received:	29th May 25
Accepted:	5th Jun 25

Transaction Report:

Dear Dr. Gorna

Thank you for the submission of your research manuscript to our journal. I apologize for the delay in handling your manuscript, but we have meanwhile received the three enclosed referee reports on it.

As you will see, the referees acknowledge the development of CLIPPERS as a potentially useful tool. That said, the referees also raise important concerns. You test and validate CLIPPERS on a single target, GroEL, and the applicability of the degrader tool beyond GroEL remains unclear at this stage. A second important concern relates to the fact that GroEL is only partially degraded and that the reduced protein levels of GroEL and the corresponding changes on protein homeostasis might not necessarily be caused by GroTAC- mediated GroEL degradation but by changes in transcription and translation due to cellular stress. These concerns regarding effectiveness, specificity and broader applicability of the method preclude publication in EMBO Reports at this stage. A validation of CLIPPERS on a second target and clear evidence for GroEL degradation caused by GroTACs and that GroEL degradation rather than secondary effects are causative of the observed proteostasis and growth phenotypes would be required. I understand that these experiments constitute a significant time and work investment, and I therefore feel that it is not productive to call for a revised version of your manuscript at this stage.

Given the potential interest of your findings, we would, however, have no objections to consider a resubmission of the manuscript in the future if you were able to address all main concerns of the reviewers as highlighted above and in their reports. I would like to stress though that such a manuscript would be treated as a new submission and would be evaluated again, also with respect to the literature and the novelty of your findings at the time of resubmission. In case you already have data at hand that can address these concerns or if you wish to discuss the reports and required experiments further, please do not hesitate to contact me.

I apologize that I cannot be more positive at this point. I hope, however, that the referee comments are going to be helpful in strengthening your indeed very interesting initial observations and I will be happy to discuss any additional data on this topic with you in the future.

Yours sincerely

Referee #1:

The manuscript by Izert-Nowakowska et al. entitled "Towards Targeted Protein Degradation in Escherichia coli - depletion of the essential GroEL protein using CLIPPERS" developed a new degradation strategy of protein targets using the highly conserved ClpXP protease. The developed system could therefore be transferred to bacteria other than the strains used. The degradation studies were supported by binding, proteomics and structural studies using cryo-EM.

The development of targeted bacterial protein degradation is challenging because bacteria, with the exception of a few strains such as Actinobacteria, lack a ubiquitin system. The authors chose to express plasmid-based degrader constructs, called Clp-interacting peptidic protein erasers (CLIPPERS), targeting the essential bacterial chaperone GroEL. CLIPPERS consisted of a peptide interacting with the protease of interest, a flexible linker, and a protein of interest binding moiety. The CLIPPER design was rationalized using structural models of ClpXP and GroEL. The GroEL binding ligand was based on a known peptide ligand, the strong binding peptide (SBP), which binds to GroEL without interfering with its activity. The reduction of GroEL levels was monitored by Western blots, and even a modest reduction of GroEL levels led to a significant reduction in growth and, as shown by MS, had a significant effect on the bacterial proteome. Proteomics showed that the number of degraded proteins increased over time (looking at 1, 2 and 6 hours of CLIPPER expression).

This is an interesting approach but my concern is that a) it would be limited to GroEL and potentially systems already involved in proteo-homeostasis and as the observed large effects on protein folding stress is not likely to happen with other targets b) the target GroEL is not significantly degraded in the proteomic study and the Western blots are according to the processed intensity data not convincing. GroEL seems therefore not the most straight forward example for developing a degrader strategy as the protein is highly regulated by heat and other stresses. Small differences that have been observed could be due to differences in transcription of translation of the target and not its degradation. c) the proposed system requires that potent peptide based ligands are known for the protein of interest that is targeted for degradation. I think therefore that the method will not be useful for most bacterial proteins. While the effects of interfering with GroEL function by this CLIPPER is interesting, I suggest the authors select an easier target protein as a first example of a CLIPPER design which will allow more mechanistic insights of the degradation process. Ideally the target would not cause toxicity and protein aggregation after interfering with its functions as these effects cause significant problems with data interpretation. Also direct monitoring (eg an activity of the target) would be an

advantage.

Referee #2:

The authors developed a novel method for targeted protein degradation in bacteria based on Clp-protease interacting peptide. Using GroEL binding peptide, authors developed GroEL-targeting CLIPPERS (Gro-TAC) to show the mode of action of the peptide degraders. The results showed that Gro-TAC successfully induced the degradation of GroEL. Furthermore, we showed that targeted degradation of GroEL significantly altered cellular proteostasis and influenced bacterial survival. These results suggest that the new methods they have developed can be used to study the function of proteins of interest and to further expand antibiotic development. The paper is well written and logically ordered. The methods section is comprehensive and clear.

I think this paper provides an important contribution to the targeted protein degradation field by developing selective bacterial protein depletion tools and providing a foundation for future studies.

The following suggestions will help the authors to improve the manuscript:

- (1) How do authors confirm the expression of peptide-base degraders?
- (2) In figure 3 G, the authors only show quantified data for GroEL levels. Please show a representative western blot result.
- (3) In figure 3H, I would like to suggest that the authors show western blots of representative up- and down-regulated proteins.
- (4) In Figure 2, GroTac2 seems to be more efficient than GroTac1 in terms of degradation efficiency and cell viability. Does GroTac2 have a stronger binding affinity than GroTac1? Or does GroTac2 form a more stable triple complex than GroTac1?
- (5) In figure 4, the authors used HIS-SUMO-peptides to measure protein and peptide affinity. The SUMO tag is relatively large for studying protein-small peptide interactions. Non-tagged peptides would be preferable for this assay.

Minor comment

- (6) In Figure S5, the addition of GroTacs decreases the amount of total intracellular protein and enolase. It is likely that the total protein level is reduced by cell death and that GroEL is degraded. Therefore, GroEL levels are either decreased due to cell death, or GroEL is genuinely degraded. How does the author interpret this?

Referee #3:

Rev

The paper

Towards Targeted Protein Degradation in Escherichia coli - depletion of the essential GroEL protein using CLIPPERS by Izert-Nowakowska et al, investigates a novel way of inducing target protein degradation by proximity in line with the implementation of degraders such as the PROTAC system in eukaryotes the the BACprotac on gram positive bacteria that possess the ClpCP system. The authors create and show the action of small peptides called CLIPPERS (Clp-interacting Peptidic Protein Erasers) to target degradation of specific proteins by the ClpXP system, so possible for both gram positive and gram negative bacteria. In particular the authors show in details the modulation of GroEL via the usage of a specific CLIPPER GroTAC and the effect on the general cell proteostasis upon even partial GroEL depletion.

The article is very well written, simple to follow. It contains all the information needed to understand the rationals and the experiments. The authors refer to the existing literature quite extensively, even though I would suggest that they should refer to the paper by McGinness et al (ref 48) earlier in the text.

The limitation, not being a problem though for the soundness of this paper and proof-of-concept about CLIPPERS, is the fact that the peptide for targeting specific degradation via proximity, still has to be expressed in a plasmid. Can the authors speculate on possible ways to deliver these peptides? In the text there is mention and reference to the difficulties related to the delivery of peptides, but this is not further argued in the discussion part. In any case, expression the CLIPPERS in an inducible plasmid is very useful for biotechnology applications. For that, it would have been nice, but not necessary for this paper, to prove the effect of another CLIPPER bound towards a different target than GroEL.

When it comes to the cryo-EM parts, I have some minor comments that are meant to possibly help with the map and model building:

- A variant of Fig4g could be made using the software OccuPy (<https://www.nature.com/articles/s41467-023-41478-1>) for the display of the densities. It will allow to show the high and low resolution part at the same time at their best, thus giving a better idea of the quality of the reconstruction and of the available density of the peptides.
- Why did the authors rely so much on ModelAngelo to build a GroEL map at a good resolution? As they finally say the map does not diverge substantially from available crystal structures.
- I would be important to try and build the peptides or to predict their binding via e.g. AF2? Or AF3 now?
- Could be beneficial to at least see/mention an asymmetric reconstruction? With no symmetry applied or with some symmetry expansion and then classification without alignment it might have been possible to build the peptides.

- The map could have been post processed with other software such as deepEMhancer, EMReady, Phenix autosharpening. The deposited map doesn't seem to be the best possible one.

On the other hand, the cryo-EM analysis is just a tiny part of the whole work, so it is not so important for the authors to answer these points.

I suggest publications of the article pretty much as it is. Would appreciate comments and answers to the points above from the authors.

** As a service to authors, EMBO Press provides authors with the ability to transfer a manuscript that one journal cannot offer to publish to another journal, without the author having to upload the manuscript data again. To transfer your manuscript to another EMBO Press journal using this service, please click on Link Not Available

Dear Dr Rembold,

We would like to urge you to reconsider your decision on Manuscript EMBOR-2024-59390-T for the following reasons.

2 out of 3 reviewers were positive and had no major objections - Reviewer 3 recommended “accept as is” whereas Reviewer 2 invited mostly additional interpretation/discussion. We are happy to revise our manuscript according to their suggestions.

It is therefore the skepticism of Reviewer 1 that we would like to address. While we believe their comments to be accurate, they exceed the standards normal for the TPD field, especially for early PROTAC prototypes. For example, the founding BacPROTAC paper (Morreale et al, Cell 2022) achieved at maximum BacPROTAC concentration (100 μ M) at most 60% reduction of the target ***fusion*** protein. They demonstrated only one single working BacPROTAC-3 prototype by qMS, achieving less than two-fold reduction of the (again, fusion) target; moreover, the authors do not provide any qMS data for Bait or Anchor only controls. The follow-up Homo-BacPROTACs (Hoi et al, Cell 2023) were able to reduce the levels of ClpC1 and ClpC2 only up to a similar range – 40% and 45-60%, respectively. We therefore consider our GroTAC example as the equally valid and logical next step to complement the possibilities opened by CymA-based BacPROTAC - of which only a couple working examples are available, with apparent similar efficiency to CLIPPERS.

We believe that we have provided the key experiment to relieve the biggest doubt of Reviewer 1 – that the GroTAC mechanism indeed relies on degradation – by the inclusion of Bait only and Anchor only controls in quantitative MS, the ultimate proof in TPD. At early time points, the controls have no effect, when GroTAC already starts acting on GroEL. Degradation-based action is also supported by the rescue of growth phenotypes in ClpP knockdowns.

The GroTAC case far outperforms anything else we have tried, is robust, reproducible, and all-round convincing using multiple controls and methods. We therefore believe it sufficient as our proof-of-concept cornerstone of the new TPD approach. We have attempted to perform additional experiments/examples, however the results were not to our satisfaction in terms of robustness, as we have seen (likely technical, but also possibly biological) variability in qMS. The cost of further qMS experiments is simply too prohibitory for us to keep playing with further variants.

By the number of requests we receive from colleagues in the field interested in our preprint, we believe that our work will be very useful and highly cited. We would like to stay with EMBO Reports as the home for our work.

Looking forward to your reply,

Maria Górna and Co-authors

Dear Dr. Gorna

Thank you for your e-mail and please apologize my delayed response. I have now had the chance to read your e-mail, re-read the referee reports and your manuscript and related studies. I do recognize the interest in bacterial degraders and also the supportive comments from referee 2 and 3.

I however also note the valid concerns from referee 1 regarding secondary effects on the proteome caused by GroEL depletion, the limited applicability of the method beyond GroEL and proteins for which a potent peptide based ligand is known at this stage of the analysis.

That said, I also understand the technical and experimental limitations and would be willing to consider a revised manuscript as a "proof-of-principle" study, which is clearly presented as such. The revised manuscript would need to clearly and transparently discuss the limitations outlined by Referee 1 and provide a careful and balanced discussion of the confounding factors using GroEL as bait. All concerns from Referee 2 and 3 need to be addressed, also in a point-by-point response.

Please note that your manuscript will be considered as a new submission in the sense that we will have to assess novelty again, taking into account any literature at the time of re-submission. If this assessment is positive, I will contact the same referees as before.

Please do not hesitate in case you have any questions.

Yours sincerely

Referee #1:

The manuscript by Izert-Nowakowska et al. entitled "Towards Targeted Protein Degradation in Escherichia coli - depletion of the essential GroEL protein using CLIPPERS" developed a new degradation strategy of protein targets using the highly conserved ClpXP protease. The developed system could therefore be transferred to bacteria other than the strains used. The degradation studies were supported by binding, proteomics and structural studies using cryo-EM. The development of targeted bacterial protein degradation is challenging because bacteria, with the exception of a few strains such as Actinobacteria, lack a ubiquitin system. The authors chose to express plasmid-based degrader constructs, called Clp-interacting peptidic protein erasers (CLIPPERS), targeting the essential bacterial chaperone GroEL. CLIPPERS consisted of a peptide interacting with the protease of interest, a flexible linker, and a protein of interest binding moiety. The CLIPPER design was rationalized using structural models of ClpXP and GroEL. The GroEL binding ligand was based on a known peptide ligand, the strong binding peptide (SBP), which binds to GroEL without interfering with its activity. The reduction of GroEL levels was monitored by Western blots, and even a modest reduction of GroEL levels led to a significant reduction in growth and, as shown by MS, had a significant effect on the bacterial proteome. Proteomics showed that the number of degraded proteins increased over time (looking at 1, 2 and 6 hours of CLIPPER expression).

Answer: We thank the Reviewer for their appreciation of the difficulty of our task to develop a new TPD approach. We agree with the Reviewer's suggestions, and we have addressed some of them in the expanded Discussion, while some are beyond the scope of this first demonstration of CLIPPER action.

This is an interesting approach but my concern is that

a) it would be limited to GroEL and potentially systems already involved in proteo-homeostasis and as the observed large effects on protein folding stress is not likely to happen with other targets

Answer ad (A)

We don't consider CLIPPERS as limited to targets involved in proteostasis. Our proof of concept regards removing an endogenous, untagged protein (where we consider our method to be superior to targeting fusion proteins) - however, not in every case this would result in a proteotoxic stress. For demonstration purposes we have chosen a conditionally essential protein, GroEL, but its involvement in proteostasis is not the requirement for targeted protein degradation per se. As mentioned by the Reviewer, any protein for which a suitable peptide ligand is known/can be developed, would be a potential target for the CLIPPER approach.

b) the target GroEL is not significantly degraded in the proteomic study and the Western blots are according to the processed intensity data not convincing. GroEL seems therefore not the most straight forward example for developing a degrader strategy as the protein is highly regulated by heat and other stresses. Small differences that have been observed could be due to differences in transcription of translation of the target and not its degradation.

Answer ad (B)

While we agree that we observed only modest effects in degradation, our work meets or even exceeds the standards normal for the TPD field, especially for early PROTAC prototypes. For example, the founding BacPROTAC paper (Morreale et al, Cell 2022) achieved at maximum BacPROTAC concentration (100 uM) at most 60% reduction of the target *fusion* protein. They demonstrated only one single working BacPROTAC-3 prototype by qMS, achieving less than two-fold reduction of the (again, fusion) target; moreover, the authors do not provide any qMS data for Bait or Anchor only controls. The follow-up Homo-BacPROTACs (Hoi et al, Cell 2023) were able to reduce the levels of ClpC1 and ClpC2 only up to a similar range – 40% and 45-60%, respectively – after a 24h treatment. We therefore consider our GroTAC example as the equally valid and logical next step to complement the possibilities opened by CymA-based BacPROTAC - of which only a couple working examples are available, with apparent similar efficiency to CLIPPERS. We added more explanations in the Discussion section to address such limitations of early PROTAC prototypes.

We believe that the strongest argument that the GroTAC mechanism indeed relies on degradation is provided by the inclusion of Bait only and Anchor only controls in quantitative MS, the ultimate proof in TPD. At early time points, the controls have no effect, when GroTAC already starts acting on GroEL. Degradation-based action is also supported by the rescue of growth phenotypes in ClpP knockdowns.

The GroTAC case far outperforms anything else we have tried, is robust, reproducible, and all-around convincing using multiple controls and methods. We therefore believe it sufficient as our proof-of-concept cornerstone of the new TPD approach. We have attempted to perform additional experiments/examples, however the results were not to our satisfaction in terms of robustness, as we have seen (likely technical, but also possibly biological) variability in qMS. The cost of further qMS experiments is simply too prohibitory for us, and in our answer to another point (C part 2) we explain the downsides of looking for further targets.

C) the proposed system requires that potent peptide based ligands are known for the protein of interest that is targeted for degradation. I think therefore that the method will not be useful for most bacterial proteins.

Answer ad (C part 1)

We agree that the CLIPPER approach requires a ligand that binds the target. This is however a requirement for every known TPD agent. We cannot therefore avoid this criticism, but it is in no way specific to our work, but rather the whole field. We do think that in many cases it might be easier to develop a peptide ligand (as was done via phage display for GroEL) than a small molecule, making CLIPPERS complementary to classical PROTACs. The important development in our work is the demonstration of the XB peptide as a suitable anchor - the ligand for ClpXP, which does not need to be redeveloped for future CLIPPERS.

C cont.) While the effects of interfering with GroEL function by this CLIPPER is interesting, I suggest the authors select an easier target protein as a first example of a CLIPPER design which will allow more mechanistic insights of the degradation process. Ideally the target would not cause

toxicity and protein aggregation after interfering with its functions as these effects cause significant problems with data interpretation. Also direct monitoring (eg an activity of the target) would be an advantage.

Answer ad (C part 2)

We agree with the Reviewer's suggestion that we should look for a target with an activity that can be followed in time. This is indeed the case of GroEL - we assayed its ATPase activity in vitro and followed its requirement for proteostasis in vivo. Screening for other targets would require either de novo development of peptide ligands (for those proteins which are known as useful reporters but have no available peptide ligands, e.g. beta lactamase) or - for neutral targets - expensive proteomics-based screening where no phenotype on cell growth/death is visible (since we would like to avoid fusion proteins appended with tags or reporters). In addition, we expect that targeting most endogenously-expressed (and therefore currently useful to bacteria) proteins, and especially all essential ones, will result in very quick follow-up changes in the bacterial metabolism (both due to bacterial adaptation and dysregulation). In due time the effect will be visible on many levels in parallel: transcription, translation and degradation, without much chance to disambiguate the order of events. Choosing a target with no obfuscating downstream effects is therefore not as easy as it seems. We therefore consider the working example of GroEL, which meets our criteria of a known ligand and a strong activity readout, as sufficient for the purpose of our proof-of-concept study.

Referee #2:

The authors developed a novel method for targeted protein degradation in bacteria based on Clp-protease interacting peptide. Using GroEL binding peptide, authors developed GroEL-targeting CLIPPERS (Gro-TAC) to show the mode of action of the peptide degraders. The results showed that Gro-TAC successfully induced the degradation of GroEL. Furthermore, we showed that targeted degradation of GroEL significantly altered cellular proteostasis and influenced bacterial survival. These results suggest that the new methods they have developed can be used to study the function of proteins of interest and to further expand antibiotic development. The paper is well written and logically ordered. The methods section is comprehensive and clear.

I think this paper provides an important contribution to the targeted protein degradation field by developing selective bacterial protein depletion tools and providing a foundation for future studies.

Answer:

We thank the Reviewer for the appreciation of the relevance of our new method for the now-only-starting bacterial TPD field, and for the useful mechanistic insights into our work. We have included some additional experiments as requested, and added more explanations to the discussion section. We apologise for the lack of antibodies in our hands to add to some of the analyses.

The following suggestions will help the authors to improve the manuscript:

(1) How do authors confirm the expression of peptide-base degraders?

Answer ad (1):

Indeed we observe peptides corresponding to our GroTACs in the quantitative TMT-MS results, and these are included in the Figure 3. All our attempts at traditional Western blot analyses with anti-Myc antibodies of these small peptides have failed, as these are tricky likely both due to the small size and large charge. A strong argument for successful expression of the peptides are the effects on phenotypes seen in Fig.2, including titratable, dose-responsive effects in Fig.2E.

(2) In figure 3 G, the authors only show quantified data for GroEL levels. Please show a representative western blot result.

Answer ad (2):

We have included the Western blot results (quantified in Figure 2G) in Supplementary Figure S5. In addition, we have now included another representative Western blot example (side-by-side replicates for the final timepoint) in the main manuscript as Figure 2H. Figure 3G represents quantification from TMT-MS data.

(3) In figure 3H, I would like to suggest that the authors show western blots of representative up- and down-regulated proteins.

Answer ad (3):

We rely on TMT-MS as a global and more quantitative measure of protein levels. The TMT-MS results are also reinforced by the enrichment analysis of whole protein groups (Supplementary Figure S7) rather than individual proteins, which helps avoid artifacts that may concern single protein levels. We do not rely on Figure 3H for our general conclusions, rather to highlight selected interesting cases from the global proteomics study. In addition, we have searched for commercially available antibodies against E. coli proteins, but couldn't find any beyond the ones we used in the study. Commercial antibodies are not routinely available for bacterial proteins, so that we would have to obtain these reagents from multiple other laboratories.

(4) In Figure 2, GroTac2 seems to be more efficient than GroTac1 in terms of degradation efficiency and cell viability. Does GroTac2 have a stronger binding affinity than GroTac1? Or does GroTac2 form a more stable triple complex than GroTac1?

Answer ad (4):

We thank the Reviewer for making a valid point. We added binary and ternary complex measurements for GroTAC1 and GroTAC2 using immobilized ZBD (as a hexamer of full-length ClpX is known to be poorly stable/too dynamic, and we cannot rely on the popular hexameric fusion of ClpX core without the necessary ZBD). Indeed, the affinities were comparable in vitro, so that we suspect that additional factors must play a role in vivo, possibly by improving steric constraints in facilitation of degradation. We included discussion of these factors.

(5) In figure 4, the authors used HIS-SUMO-peptides to measure protein and peptide affinity. The SUMO tag is relatively large for studying protein-small peptide interactions. Non-tagged peptides would be preferable for this assay.

Answer ad (5):

We thank the Reviewer for this suggestion, indeed for a peptide the size of SUMO might be relatively large. In the initial assays, we used a His-SUMO tagged GroTAC (and controls) for binary affinities with untagged ClpX and GroEL. Since the hexameric form of full-length ClpX is poorly stable/too dynamic in vitro, we switched to using His-tagged ZBD dimer as the immobilized ligand in BLI assays. This allowed us to measure binary and even ternary binding constants using untagged GroTACs and control peptides. The small size of ZBD might have enabled the BLI signal measurements with such small peptides. The results for binary complexes remained comparable to those obtained with His-SUMO-peptides and the full-length ClpX as the analyte. We are therefore happy that switching the tagged components in the binary system has demonstrated no impact of the N-terminal tag on GroTAC binding.

To assess the influence of the His-SUMO tag and to compare GroTAC1 & 2 we have very much improved our binding study - we used ZBD to show binding of untagged synthetic peptides and controls. We achieved even the hallmark of TPD studies, i.e. we showed ternary complex formation (ZBD-GroTAC-GroEL) and compared the so-called “ternary affinities” of GroTAC1 and GroTAC2. We are therefore very grateful to the helpful suggestions of the Reviewer that prompted us to improve our study.

Minor comment

(6) In Figure S5, the addition of GroTacs decreases the amount of total intracellular protein and enolase. It is likely that the total protein level is reduced by cell death and that GroEL is degraded. Therefore, GroEL levels are either decreased due to cell death, or GroEL is genuinely degraded. How does the author interpret this?

Answer ad (6):

We agree that at later time points, the bactericidal effect of GroTACs already takes place and protein production is globally affected. TMT-MS analyses at 6h show however, that in the 1/3 of the affected proteome, there are more proteins significantly upregulated than downregulated (819 and 519, respectively). The GroEL levels are already affected at earlier timepoints (Fig.3G). We assume that further GroEL degradation goes in hand with the progressive functional effect of GroEL knockdown on the proteome, and that it cannot be easily separated in time. It is worth noting, that in *M. tuberculosis*, similar phenotypes and protein reduction in ClpC1 and ClpC2 was reported after 24h (Hoi et al Cell 2023), but *E.coli* has much faster generation time and the effects we observe take place sooner. We added discussion on the comparison on the metabolic rates of the different bacteria.

Referee #3:

The paper “Towards Targeted Protein Degradation in *Escherichia coli* - depletion of the essential GroEL protein using CLIPPERS” by Izert-Nowakowska et al, investigates a novel way of inducing target protein degradation by proximity in line with the implementation of degraders such as the PROTAC system in eukaryotes the the BACprotac on gram positive bacteria that possess the

ClpCP system. The authors create and show the action of small peptides called CLIPPERS (Clp-interacting Peptidic Protein Erasers) to target degradation of specific proteins by the ClpXP system, so possible for both gram positive and gram negative bacteria. In particular the authors show in details the modulation of GroEL via the usage of a specific CLIPPER GroTAC and the effect on the general cell proteostasis upon even partial GroEL depletion.

The article is very well written, simple to follow. It contains all the information needed to understand the rationals and the experiments. The authors refer to the existing literature quite extensively, even though I would suggest that they should refer to the paper by McGinness et al (ref 48) earlier in the text.

The limitation, not being a problem though for the soundness of this paper and proof-of-concept about CLIPPERS, is the fact that the peptide for targeting specific degradation via proximity, still has to be expressed in a plasmid. Can the authors speculate on possible ways to deliver these peptides? In the text there is mention and reference to the difficulties related to the delivery of peptides, but this is not further argued in the discussion part. In any case, expression the CLIPPERS in an inducible plasmid is very useful for biotechnology applications. For that, it would have been nice, but not necessary for this paper, to prove the effect of another CLIPPER bound towards a different target than GroEL.

Answer:

We thank the Reviewer for the succinct analysis of the limitations of peptides as drugs and pointing out the usefulness of our plasmid-based approach for biotechnology. We indeed focus on GroTACs and GroEL as the handy example that serves as the proof of concept, and leave further more extensive screening for future work (as was also done for cymA-based BacPROTACs and HomoBacPROTACs). We have now included and explained the use of the tools mentioned by the Reviewer, and added some additional discussion on peptide delivery.

When it comes to the cryo-EM parts, I have some minor comments that are meant to possibly help with the map and model building:

- A variant of Fig 4G could be made using the software OccuPy (<https://www.nature.com/articles/s41467-023-41478-1>) for the display of the densities. It will allow to show the high and low resolution part at the same time at their best, thus giving a better idea of the quality of the reconstruction and of the available density of the peptides.

Answer:

Thank you for your suggestion, we included the OccuPy-generated analysis in Figure 4H in the manuscript.

- Why did the authors rely so much on ModelAngelo to build a GroEL map at a good resolution? As they finally say the map does not diverge substantially from available crystal structures.

Answer:

Thank you, that is a reasonable question. As you point out, the structure of GroEL is well-known and readily available, so the most important and novel part of our work was to build the model of interaction between our peptide and GroEL. You are right to point out that the map is at a good resolution, but as we hoped to demonstrate in Fig. 4 mentioned above, the map has significantly worse resolution around the most important part - the interface between GroEL and peptide. In the beginning of the process, we had different hypotheses of how the peptide binds to GroEL (one of them mentioned in the answer to your question about AlphaFold), so we wanted to build the model of the peptide using an unbiased *de novo* method. Out of many methods we tested, only ModelAngelo treated the density near the expected peptide binding site as occupied by atoms. Therefore, we relied heavily on the model, the positions of CA atoms, and the probabilities of individual amino acids from ModelAngelo to build multiple peptide models as hypotheses. After refinements and testing these hypotheses of peptide binding to GroEL, we concluded that the most probable structure is very similar to the available crystal structure. However, because we tried to construct the peptide model from scratch, we relied on ModelAngelo, especially in the beginning.

- I would be important to try and build the peptides or to predict their binding via e.g. AF2? Or AF3 now?

Answer:

Thank you for your valuable suggestion. We used AF2 to predict the structure of peptide and its binding to the GroEL interface. In the most reliable model we obtained (reliable by AF metrics such as pLLDT, PAE, and ipTM), AlphaFold predicted that the XB part of the peptide was bound to GroEL, instead of the expected SBP part. We ran some additional (wet lab) experiments which didn't show any binding of only the XB part of the peptide to GroEL. Additionally, after careful examination of the cryoEM map, we did not find any further evidence that the model predicted by AF2 was correct. We omitted these results from the manuscript for clarity purposes. After your suggestion we tried the new AF3 and this time the prediction was very similar to the model we have reconstructed from the cryoEM map - since it did not change our conclusions, we relied on our initial reconstruction.

- Could be beneficial to at least see/mention an asymmetric reconstruction? With no symmetry applied or with some symmetry expansion and then classification without alignment it might have been possible to build the peptides.

Answer:

Thank you for your suggestion. We attempted to build the peptides using an asymmetric map, but it did not result in a clearer picture of the density around the peptide nor better peptide models. We omitted these results from the manuscript for clarity purposes. We have added a mention of the asymmetric reconstruction to the methods section.

- The map could have been post processed with other software such as deepEMhancer, EMReady, Phenix autosharpening. The deposited map doesn't seem to be the best possible one.

Answer:

Thank you for your suggestion. We tried post-processing the map in multiple ways, but almost all of them improved the quality of the map around GroEL and simultaneously made it even harder to build models of the peptide. As the peptide is the most important part of our model, we eventually decided to deposit the map where the density around the peptide is most clearly seen.

On the other hand, the cryo-EM analysis is just a tiny part of the whole work, so it is not so important for the authors to answer these points.

I suggest publications of the article pretty much as it is. Would appreciate comments and answers to the points above from the authors.

Answer:

We thank the Reviewer for their useful suggestions and we are happy our work meets the Reviewer's overall expectations.

Dear Dr. Gorna

Thank you for the submission of your revised manuscript to EMBO Reports. We have now received the reports from the referees that were asked to assess it (copied below). As you will see, both referees conclude that the revision has adequately addressed their concerns and consider your work suitable for publication as a proof-of-principle study that will stimulate further work on bacterial TPDs.

Before I can officially accept your manuscript for publication in EMBO Reports, I need you to format your manuscript according to our guidelines. You find the general instructions below, but in order to streamline and expedite the process, I list a few specific points below.

Once we have received your revised manuscript, we will perform a number of quality control steps, including a routine data integrity check. A data editor will check your figure legends and Data availability section, whether n, statistical tests, exact p-values etc have been specified and a link to the datasets has been provided, respectively.

We will also need source data, i.e., minimally modified raw data. You will receive a separate e-mail with instructions.

1) Specific comments:

- Please describe your findings in the abstract in present tense.
- Please do not use hyphens in titles.
- Please format the references in an alphabetical manner.
- Your study will be published in our Reports section. To comply with this format, I kindly ask you to combine Results and Discussion.
- Please order the manuscript sections like this:
Title page - Abstract - Introduction - Results and Discussion - Methods - Acknowledgements - Disclosure and competing interests statement - References - Figure legends - Tables and their legends (not EV tables) - Expanded View Figure legends
- Access to proteomics (PXD045730) and structural data (PDB ID 8S32) must be listed in a dedicated Data availability section, with an URL that links directly to the dataset. See point (7) below.
- All figure legends need a definition of "n" and whether it is technical or biological replicates. Bars and error bars need to be defined. The individual datapoints should be shown. Exact p-values need to be reported. See point (10) in the general instructions.
- Please provide a Reagents and Tools table (see point 12).
- The Conflict of interest is called "Disclosure and competing interests statement".
- Regarding the Author Contributions, we now use CRediT to specify the contributions of each author in the journal submission system. Therefore, please remove the Author Contributions from the manuscript file and make sure that the author contributions in our online manuscript tracking system are correct and up-to-date. The information you specified in the system will be automatically retrieved and typeset into the article. You can enter additional information in the free text box provided, if you wish.
- The nomenclature for Supplementary Information is Appendix with Appendix Figure S# and Appendix Table S#. Some of the tables might be part of the Reagents and Tools table, such as Table S4. The Appendix needs a title page with a table of content incl. page numbers.
- Table S5 should be Dataset EV1, reflecting the complex nature of the data/table.

=====

2) General formatting instructions:

When submitting your revised manuscript, we need:

2) individual production quality figure files as .eps, .tif, .jpg (one file per figure). Please download our Figure Preparation Guidelines (figure preparation pdf) from our Author Guidelines pages <https://www.embopress.org/page/journal/14693178/authorguide> for more info on how to prepare your figures.

4) a complete author checklist, which you can download from our author guidelines (<<https://www.embopress.org/page/journal/14693178/authorguide>>). Please insert information in the checklist that is also reflected in the manuscript. The completed author checklist will also be part of the RPF.

5) Please note that all corresponding authors are required to supply an ORCID ID for their name upon submission of a revised manuscript (<<https://orcid.org/>>). Please find instructions on how to link your ORCID ID to your account in our manuscript tracking system in our Author guidelines (<<https://www.embopress.org/page/journal/14693178/authorguide#authorshipguidelines>>)

6) We replaced Supplementary Information with Expanded View (EV) Figures and Tables that are collapsible/expandable online. A maximum of 5 EV Figures can be typeset. EV Figures should be cited as "Figure EV1, Figure EV2" etc... in the text and their respective legends should be included in the main text after the legends of regular figures.

7) Before submitting your revision, primary datasets (and computer code, where appropriate) produced in this study need to be deposited in an appropriate public database (see <<https://www.embopress.org/page/journal/14693178/authorguide#dataavailability>>).

The accession numbers and database should be listed in a formal "Data Availability" section (placed after Materials & Method) that follows the model below (see also <<https://www.embopress.org/page/journal/14693178/authorguide#dataavailability>>). Please note that the Data Availability Section is restricted to new primary data that are part of this study.

Data availability

Additional information on source data and instruction on how to label the files are available <<https://www.embopress.org/page/journal/14693178/authorguide#sourcedata>>

10) Figure legends and data quantification:
The following points must be specified in each figure legend:

- the name of the statistical test used to generate error bars and P values,
 - the EXACT p-values,
 - the number (n) of independent experiments (please specify technical or biological replicates) underlying each data point,
 - the nature of the bars and error bars (s.d., s.e.m.)
-
- If the data are obtained from n {less than or equal to} 5, show the individual data points in addition to the SD or SEM.
 - If the data are obtained from n {less than or equal to} 2, use scatter blots showing the individual data points.

11) Our journal encourages inclusion of *data citations in the reference list* to directly cite datasets that were re-used and obtained from public databases. Data citations in the article text are distinct from normal bibliographical citations and should directly link to the database records from which the data can be accessed. In the main text, data citations are formatted as follows: "Data ref: Smith et al, 2001" or "Data ref: NCBI Sequence Read Archive PRJNA342805, 2017". In the Reference list, data citations must be labeled with "[DATASET]". A data reference must provide the database name, accession number/identifiers and a resolvable link to the landing page from which the data can be accessed at the end of the reference. Further instructions are available at <<https://www.embopress.org/page/journal/14693178/authorguide#referencesformat>>.

12) All Materials and Methods need to be described in the main text using our 'Structured Methods' format. According to this format, the Methods section includes a Reagents and Tools Table (listing key reagents, experimental models, software and relevant equipment and including their sources and relevant identifiers) followed by a Methods and Protocols section describing the methods, ideally using a step-by-step protocol format. The aim is to facilitate adoption of the methodologies across labs. Please download and fill our Reagents and Tools Table template (.docx), which you can find in our author guidelines:

13) As part of the EMBO publication's Transparent Editorial Process, EMBO Reports publishes online a Review Process File to accompany accepted manuscripts. This File will be published in conjunction with your paper and will include the referee reports, your point-by-point response and all pertinent correspondence relating to the manuscript.

Yours sincerely,

=====

Referee #2:

The authors have addressed my concerns and thoughtfully discussed the points raised by the other reviewers. I agree with their interpretation that the early prototype of PROAC has limited impact on degradation, and I'm genuinely excited to see how the second-generation CLIPPERS will develop. This work contributes to the bacterial targeted protein degradation field.

This revised version is suitable for publication in EMBO Reports. I have just one remaining short question:

Q: In Figure 2H, the authors present the GroEL levels. Could you include loading controls?

Referee #3:

The authors took all my comments under consideration and have modified the manuscript accordingly. I think it is important to acknowledge the importance in the development of novel bacterial TPDs. The work as one clear message: a proof-of-concept of a novel TPD. All the best

All editorial and formatting issues were resolved by the authors.

Dr. Maria Gorna
University of Warsaw
Structural Biology Group, Biological and Chemical Research Centre, Faculty of Chemistry
Warsaw
Poland

Dear Dr. Gorna,

I am very pleased to accept your manuscript for publication in the next available issue of EMBO reports. Thank you for your contribution to our journal.

Yours sincerely,
